JCB Journal of Cell Biology

# Pex30-dependent membrane contact sites maintain ER lipid homeostasis

Joana Veríssimo Ferreira[1] , Yara Ahmed[2] , Tiaan Heunis[1] , Aamna Jain[4,5] , Errin Johnson[1] , Markus Räschle[6] , Robert Ernst[4,5] , Stefano Vanni[2,3] , and Pedro Carvalho[1]

In eukaryotic cells, communication between organelles and the coordination of their activities depend on membrane contact sites (MCS). How MCS are regulated under the dynamic cellular environment remains poorly understood. Here, we investigate how Pex30, a membrane protein localized to the endoplasmic reticulum (ER), regulates multiple MCS in budding yeast. We show that Pex30 is critical for the integrity of ER MCS with peroxisomes and vacuoles. This requires the dysferlin (DysF) domain on the Pex30 cytosolic tail. This domain binds to phosphatidic acid (PA) both in vitro and in silico, and it is important for normal PA metabolism in vivo. The DysF domain is evolutionarily conserved and may play a general role in PA homeostasis across eukaryotes. We further show that the ER–vacuole MCS requires a Pex30 C-terminal domain of unknown function and that its activity is controlled by phosphorylation in response to metabolic cues. These findings provide new insights into the dynamic nature of MCS and their coordination with cellular metabolism.

## Introduction

The hallmark of eukaryotic cells is the presence of membrane-bound organelles with specialized functions. The coordinated activities of the various organelles, critical for cellular homeostasis, are facilitated by membrane contact sites (MCS). These are regions in which the membranes of two or more organelles come into close proximity to exchange signals and molecules (Scorrano et al., 2019). Studies in yeast and mammalian cells have revealed that most cellular organelles establish MCS, identified numerous MCS components, and demonstrated their dynamic behavior in response to cellular metabolism (Prinz et al., 2020; Voeltz et al., 2024). Consistent with their pervasive nature and importance, mutations in MCS proteins have been linked to a constellation of diseases, such as neurological diseases (Kim et al., 2022; Venditti et al., 2021).

The endoplasmic reticulum (ER), the main organelle for cellular lipid synthesis, establishes prominent MCS that serve as major gateways for lipid trafficking to other organelles such as mitochondria, peroxisomes, lysosomes, lipid droplets (LDs), and even the plasma membrane (Valm et al., 2017; Shai et al., 2018; Wu et al., 2018). While several components of ER MCS have been identified, their function and regulation at these specialized membrane regions remain ill-defined.

In this study, we focus on Pex30, an ER membrane protein that concentrates at multiple ER MCS in the budding yeast *Saccharomyces cerevisiae*. Pex30 localization to the different MCS is specified by its binding to evolutionary-related adaptor proteins that define the Pex30 family (Ferreira and Carvalho, 2021). When bound to Pex28 and Pex32, Pex30 localizes to ER MCS with peroxisomes, while the binding to Pex29 promotes Pex30 accumulation at the nucleus–vacuole junction (NVJ). Pex30 also concentrates at ER MCS with LDs, a localization that appears to be independent of any known partner proteins. The Pex30 family also includes Pex31, a less studied member that appears to function independently of Pex30 (Ferreira and Carvalho, 2021).

The role of Pex30 at MCS is not fully understood, but mutations in Pex30 or other family members result in defects in associated organelles, such as peroxisomes and LDs (Joshi et al., 2018; Wang et al., 2018). For example, mutations in Pex30 family members show defects in peroxisomal matrix protein import and cause abnormal number and size of peroxisomes (David et al., 2013; Ferreira and Carvalho, 2021; Joshi et al., 2016; Vizeacoumar et al., 2003, 2004, 2006). Mutations in Pex30 homologues in other fungi, such as *Hansenula polymorpha* or *Yarrowia lipolytica*, also resulted in peroxisome defects (Brown et al., 2000; Tam and Rachubinski, 2002; Wu et al., 2020).

[1]Sir William Dunn School of Pathology, University of Oxford, Oxford, UK;  [2]Department of Biology, University of Fribourg, Fribourg, Switzerland;  [3]Swiss National Center for Competence in Research Bio-inspired Materials, University of Fribourg, Fribourg, Switzerland;  [4]Medical Biochemistry and Molecular Biology, Saarland University, Homburg, Germany;  [5]Preclinical Center for Molecular Signaling, Saarland University, Homburg, Germany;  [6]Department of Molecular Genetics, TU Kaiserslautern, Kaiserslautern, Germany.

Correspondence to Pedro Carvalho: pedro.carvalho@path.ox.ac.uk.

Moreover, Pex30 mutants show abnormal NVJ organization, with defective distribution of LDs and impaired recruitment of NVJ components, such as Nvj1. These defects are also observed in cells lacking Pex29, the Pex30 adaptor at the NVJ (Ferreira and Carvalho, 2021).

Aside from its role at MCS, Pex30, and possibly Pex31, can also regulate other aspects of ER homeostasis. The over-expression of Pex30 or Pex31 was shown to suppress the lethality of a *rtn1Δrtn2Δyop1Δspo7Δ* mutant, which lacks key ER shaping proteins (Rtn1, Rtn2, and Yop1) and a major regulator of phosphatidic acid (PA) (Spo7), essential for lipid homeostasis (Joshi et al., 2016). This suppression has been attributed to the role of Pex30 and Pex31 in shaping ER membranes, mainly via their membrane domain called reticulon-homology domain (RHD) (Joshi et al., 2016), which is similar to the RHD found in reticulons, the prototypical ER shaping proteins (Voeltz et al., 2006). Pex30 membrane shaping activity has been also implicated in organizing ER domains for the biogenesis of peroxisomes and LDs (Choudhary et al., 2020; Joshi et al., 2018; Wang et al., 2018). Normally, proteins with RHDs function as oligomers (Shibata et al., 2008; Wang et al., 2021), and in the case of Pex30, the RHD is important for the interactions with its family members/adaptor proteins (Ferreira and Carvalho, 2021).

Here, we investigate how the other Pex30 domains, namely, the dysferlin (DysF) domain and the domain of unknown function (DUF), contribute to its function at various MCS. We show that DysF binds to PA both in vitro and in silico and that this activity is essential for PA homeostasis in vivo. In contrast to DysF, which is essential for Pex30 function at all MCS, we found that the DUF is specifically required at the NVJ and that its function at this MCS is regulated by phosphorylation. Overall, our findings suggest that Pex30 family members play a central role in ER PA metabolism and highlight the importance of MCS in ER lipid homeostasis.

## Results

### Pex30 is required for the integrity of ER MCS

We previously showed that Pex30 uses different adaptor proteins to function at distinct contact sites (Fig. 1 A) (Ferreira and Carvalho, 2021). However, the importance of Pex30 in maintaining MCS integrity remains unclear. To assess the contribution of Pex30 to MCS formation, we analyzed two of the MCS to which it localizes: the NVJ and the ER–peroxisome contacts. For the NVJ, we used volume electron microscopy (vEM), an approach that allowed us to assess the proximity between nuclear ER and the vacuolar membranes to form the NVJ (Pan et al., 2000). As previously described (Li and Kane, 2009; Hariri et al., 2018), we observed that the NVJ becomes prominent as wild-type (WT) cells transition from exponential growth toward the diauxic shift and stationary phase (Fig. 1 B and Fig. S1 A). During the diauxic shift, NVJ expansion resulted in a reduction in the distance between the nuclear ER and the vacuolar membranes (Fig. 1 C), as well as an increase in the area in which the two membranes are apposed (Videos 1 and 2). During this period, we observed that most WT cells (74%) accumulate LDs in regions adjacent to the NVJ (Fig. S1 B, Videos 1 and 2), as

previously reported (Hariri et al., 2018). These observations indicated that vEM is a suitable methodology for structural analysis of the NVJ. During the diauxic shift, cells lacking Pex30 or its NVJ adaptor Pex29 displayed an NVJ defect (Fig. 1 B), with increased distance between nuclear ER and vacuolar membranes (Fig. 1 C), and consequently a reduction in the juxtaposition between these organelles (Videos 3, 4, 5, and 6). Interestingly, these mutants also showed fragmented vacuoles (Fig. 1 D and Fig. S1 B). Similar defects were observed in cells lacking the well-characterized NVJ tether Nvj1 (Fig. 1, A–D and Fig. S1 B, Videos 7 and 8). These results indicate that the Pex30-Pex29 complex contributes to the integrity of the NVJ.

To determine the importance of Pex30 at the ER–peroxisome MCS, we monitored the localization of Inp1. Inp1 is a soluble cytosolic protein that concentrates at ER–peroxisome contacts and bridges proximal Pex3 molecules in the ER and peroxisomal membranes (Fig. 1 A) (Knoblach et al., 2013). Consistent with earlier studies, endogenous Inp1 expressed as a C-terminal mCherry fusion (Inp1-mCherry) colocalized with endogenous Pex3 tagged with mNeonGreen (Pex3-mNG) (Fig. S1 C). In WT cells, most of Inp1-mCherry foci also colocalized with Pex32 (89.27% of foci), which, together with Pex28, targets Pex30 to ER–peroxisome MCS (Fig. 1 A) (Ferreira and Carvalho, 2021). In contrast, Inp1-mCherry foci were not detected in *pex30Δ* cells (Fig. 1 E), despite Inp1 being expressed at normal levels (Fig. 1 F and Fig. S1 D), suggesting that ER–peroxisome contacts were defective and Inp1 was diffuse throughout the cytosol. In contrast, the *pex29Δ* mutant, with impaired Pex30 function at the NVJ but expected to have normal ER–peroxisome contacts, showed Inp1-mCherry foci like WT cells (Fig. 1 E). Collectively, these data indicate that Pex30 complexes are important for the integrity of the NVJ and ER–peroxisome MCS.

### Distinct requirements of Pex30 domains at different MCS

Pex30 is anchored to the ER membrane via the RHD, a domain common among ER shaping proteins and that has membrane curvature–inducing activity (Hu et al., 2008). The RHD also mediates the binding of Pex30 to the adaptor proteins that facilitate its accumulation at multiple MCS (Ferreira and Carvalho, 2021). The extended Pex30 cytosolic C-terminus contains a DysF domain and a region annotated as a domain of unknown function 4196 (hereafter called the DUF domain) (Fig. 2 A). These domains are poorly characterized but are expected to affect Pex30 differently: while *Pex30DysFΔ*, a mutant lacking the DysF domain, behaves like *pex30Δ* cells, *Pex30DUFΔ*, lacking the DUF domain, has no reported phenotype (Ferreira and Carvalho, 2021). Importantly, both *Pex30DysFΔ* and *Pex30DUFΔ* are expressed at normal levels, suggesting that they affect Pex30 function by a different mechanism (Ferreira and Carvalho, 2021). To gain further insight into the roles of the DysF and DUF domains, we analyzed their contribution to the ER–peroxisome contacts and the NVJ. To assess the integrity of ER–peroxisome contacts, we monitored the localization of Inp1, as described above. In cells expressing *Pex30DysFΔ*, Inp1-mCherry was mislocalized (Fig. 2 B). These cells also showed abnormal localization of Pex32-mNG, an adaptor that, together with Pex28, targets Pex30 to ER–peroxisome contacts (Fig. 2 B). Since the interaction of Pex30

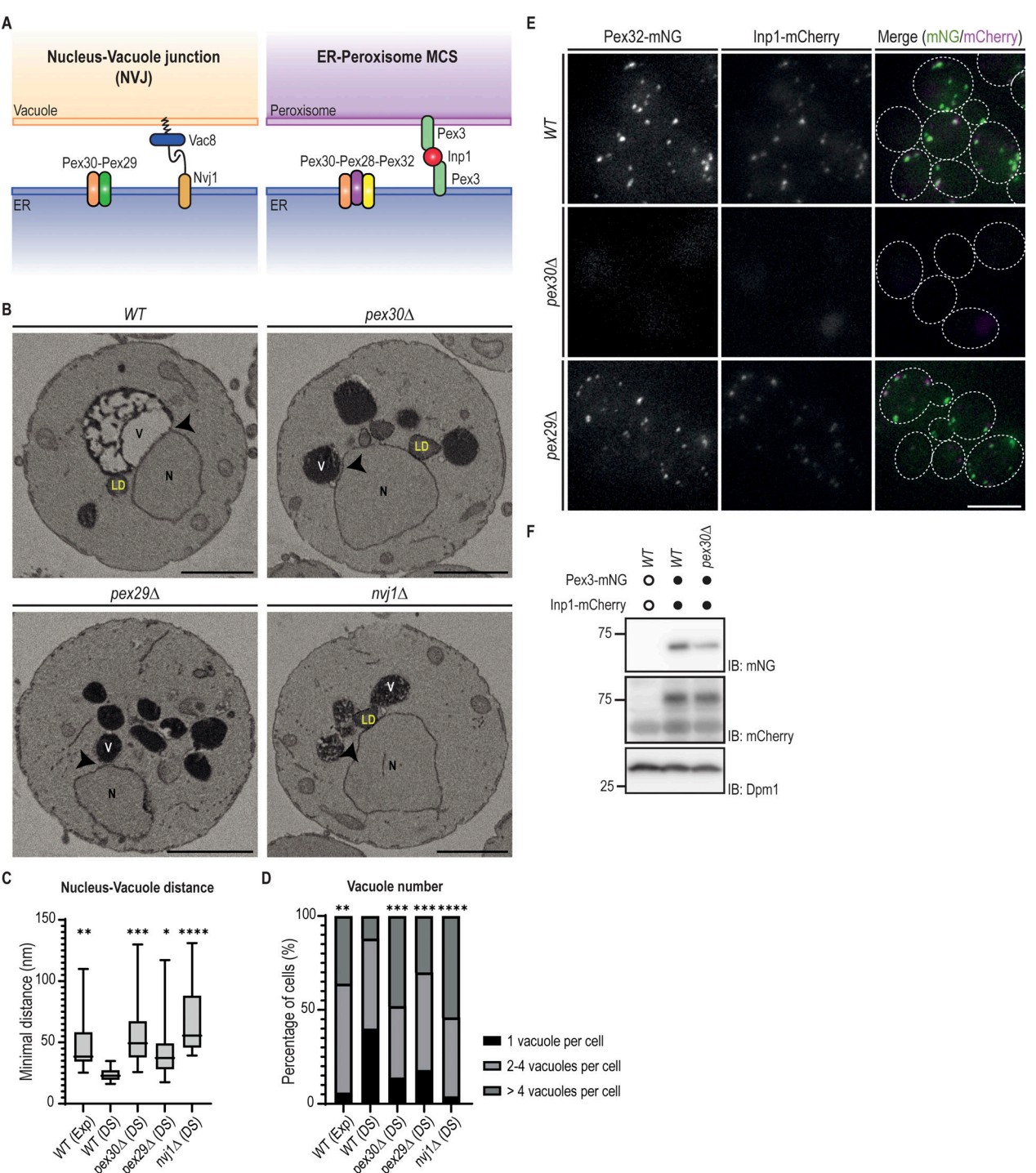

Figure 1. **Pex30 is required for the integrity of the NVJ and ER–peroxisome MCS. (A)** Schematic representation of the NVJ and the MCS between the ER and peroxisomes. Left: the Pex30–Pex29 complex accumulates at the NVJ. The binding of Nvj1 (in the ER) to Vac8 (in the vacuole) defines the prototypical NVJ tether. Right: the Pex30–Pex28-Pex32 complex accumulates at the ER–peroxisome MCS. The cytosolic protein Inp1 bridges Pex3 molecules in the ER and peroxisomes and is part of the tether between the two organelles. **(B)** Single Z-slices of *WT*, *pex30Δ*, *pex29Δ*, and *nvj1Δ* spheroplasts during the diauxic shift from volumes acquired using SBF-SEM 3View. The site of the minimal distance between the nucleus and the vacuole is indicated by an arrowhead. N, nucleus; V, vacuole; LD, lipid droplet; SBF-SEM, serial block-face scanning electron microscopy. Bars, 1 μm. **(C)** Quantification of the minimal distance between the nucleus and the closest vacuole in cells grown as in B. 20 cells per genotype and per condition were analyzed. Box and whiskers represent the distribution of the values (minimum, 25th percentile, median, 75th percentile, and maximum). Ordinary one-way ANOVA and Dunnett's multiple comparisons were used to compare the minimal distance between the organelles with the WT (DS) condition (****, P < 0.0001; ***, P < 0.001; **, P < 0.01; *, P < 0.05). **(D)** Quantification of the number of vacuoles per cell, in cells treated as in B. Percentage of cells with 1, 2–4, or >4 vacuoles is indicated for each condition. 50 cells per genotype and condition were analyzed. Ordinary one-way ANOVA and Dunnett's multiple comparisons were used to compare the percentage of cells with one vacuole with the WT (DS) condition (****, P < 0.0001; ***, P < 0.001; **, P < 0.01). **(E)** Localization of endogenous Pex32-mNG and Inp1-mCherry in exponentially growing cells with the indicated genotype. Bar, 5 μm. **(F)** Steady-state levels of endogenously tagged Pex3-mNG and Inp1-mCherry in cells with the indicated

genotype. Whole-cell extracts were prepared from exponentially growing cells, separated by SDS-PAGE, and analyzed by western blotting. Pex3-mNG, Inp1-mCherry and Dpm1, used as a loading control, were detected with anti-mNG, anti-mCherry, and anti-Dpm1 antibodies, respectively. IB, immunoblot. The position of molecular weight markers (75 and 25 kDa) is shown. Source data are available for this figure: SourceData F1.

with Pex32 and Pex28 is unaffected in $Pex30^{DysF\Delta}$ cells (Fig. S2 A), these data indicate that the DysF domain of Pex30 is required for Pex30-Pex28-Pex32 localization and function at ER–peroxisome contacts. In contrast, in $Pex30^{DUF\Delta}$ cells, the localization of both Inp1-mCherry and Pex32-mNG was indistinguishable from WT cells (Fig. 2 B). Moreover, $Pex30^{DUF\Delta}$ cells showed normal numbers of peroxisomes (Fig. 2 C), which are competent in importing the peroxisomal matrix marker mCherry-PTS1 (Fig. 2 D). In contrast, $Pex30^{DysF\Delta}$ cells have an increased number of peroxisomes as observed in $pex30\Delta$ mutants (Ferreira and Carvalho, 2021; Deori et al., 2023). Altogether, these data indicate that the DUF domain is dispensable for Pex30 function at ER–peroxisome contacts.

To analyze the contribution of DysF and DUF domains at the NVJ, we monitored Pex30 localization to this MCS, as described previously (Ferreira and Carvalho, 2021). In stationary phase cells, endogenous Pex30 localizes throughout the ER and concentrates at the NVJ, as detected by the specific marker protein Nvj1-tdTomato (Fig. 2 E). Endogenously mNG-tagged $Pex30^{DysF\Delta}$ and $Pex30^{DUF\Delta}$ displayed a similar distribution along the ER. However, both $Pex30^{DysF\Delta}$ and $Pex30^{DUF\Delta}$ failed to concentrate at the NVJ (Fig. 2 E). Interestingly, both mutants also showed an abnormal distribution of Nvj1-tdTomato to the nuclear ER (Fig. 2, E and F), suggesting a general disruption of the NVJ. Moreover, the typical clustering of LDs at the NVJ observed in WT cells was also lost in $Pex30^{DysF\Delta}$ and $Pex30^{DUF\Delta}$ cells (Fig. 2, E–F). A similar phenotype was observed in $pex30\Delta$ mutants (Ferreira and Carvalho, 2021). Therefore, the Pex30 function at the NVJ requires both its DysF and DUF domains.

## DysF is a PA-binding domain

The DysF domain is present in all Pex30 family members (Ferreira and Carvalho, 2021; Joshi et al., 2016; Vizeacoumar et al., 2006), including Pex29 and Pex32, the adaptors that facilitate Pex30 accumulation at the NVJ and ER–peroxisome contacts, respectively. To test the importance of their DysF domains for MCS integrity, we deleted Pex29 and Pex32 DysF domains. The expression of $Pex29^{DysF\Delta}$ and $Pex32^{DysF\Delta}$ resulted in abnormal localization of Nvj1-tdTomato (Fig. S2 B) and Inp1-mCherry (Fig. S2 C), respectively. As in the case of Pex30, deletion of the DysF domain did not greatly affect the levels of Pex29 and Pex32, or their interactions with Pex30 (Fig. S2 A). Thus, these data further support the notion that the DysF domains in Pex30 and its partner proteins perform critical but ill-defined functions.

Studies on other DysF domain–containing proteins, such as yeast Spo73 (Parodi et al., 2015; Okumura et al., 2015; Nakamura et al., 2021), human Dysferlin (Grounds et al., 2014; Haynes et al., 2019), and human TECPR1 (Kaur et al., 2023; Corkery et al., 2023; Boyle et al., 2023), suggested links between the DysF domain and lipid-related processes. Therefore, we tested whether the DysF domain binds lipids directly. The DysF domain of

Pex30 was recombinantly expressed and purified using affinity and size-exclusion chromatography (Fig. S3 A). The ability and specificity of purified DysF to bind lipids were analyzed using lipid strips. This assay suggested that the DysF domain has an affinity to PA (Fig. 3 A). Binding to other negatively charged phospholipids such as phosphatidylinositol 4-phosphate was also observed, but to a much lower extent. A similar binding pattern was observed for the Opi1 amphipathic helix, a well-characterized PA-binding domain (Hofbauer et al., 2018; Loewen et al., 2004), and for the purified DysF domains of Pex28, Pex29, Pex31, as well as the innerDysF domain of human DysF (Fig. S3 B). We were unable to analyze the DysF domain of Pex32, as this protein was prone to aggregation and recalcitrant to purification. Similar results were obtained when lipid-binding capacity of DysF was assessed by a liposome flotation assay (Fig. 3 B), using the peptide His-Sumo as a control (Fig. S3 C). The DysF domains of Pex30 (Fig. 3, B and C), Pex31 (Fig. S3 D), and human Dysferlin (Fig. S3 E) interacted with liposomes containing 40% PA but not with equivalent levels of phosphatidylserine (PS). In contrast, no significant interaction with the DysF domain of Pex30 and liposomes containing 5% PI(4)P was detected (Fig. 3, B and C). Altogether, these data indicate that DysF is a lipid-binding domain with an affinity for PA.

To characterize the binding mode of the Pex30 DysF domain to PA-rich membranes, we employed coarse-grained molecular dynamics (CG-MD) simulations using MARTINI 3 (Souza et al., 2021) (Fig. 4, A–C), as this protocol has been shown to identify the correct binding interface for membrane-binding peripheral proteins (Srinivasan et al., 2021). Unbiased CG-MD simulations in which the Pex30 DysF domain is initially placed far away from membranes containing varying amounts of phosphatidyl-choline (PC) and PA reproduce the in vitro observation, with the DysF domain binding to membranes as their PA content increases (Fig. 4 A). Control simulations in which PA was replaced by phosphatidylethanolamine (PE) resulted in no binding, whereas similar binding of DysF was observed when PA was replaced by PS (Fig. S3 F). However, in all-atom (AA) simulations started from back-mapped conformations of the DysF domain bound to PA-rich bilayers, the binding mode of DysF to the bilayer remained constant throughout the trajectory (Fig. S3 G); on the contrary, in the presence of PS, the binding interface was not maintained over time (Fig. S3 G). These results indicate that the PS binding observed in CG simulations is likely to originate from intrinsic inaccuracies of our CG protocol, which was already shown not to be well-suited to discriminate between lipids with identical net charge (Srinivasan et al., 2021, 2024). Characterization of the binding interface in the simulations with 40% PA identified two main membrane-binding regions, in particular those involving residues 296–304 and residues 378–395 (Fig. 4, B and C). The regions involved in lipid interactions are enriched in aromatic (W298, W303, F392, Y395)

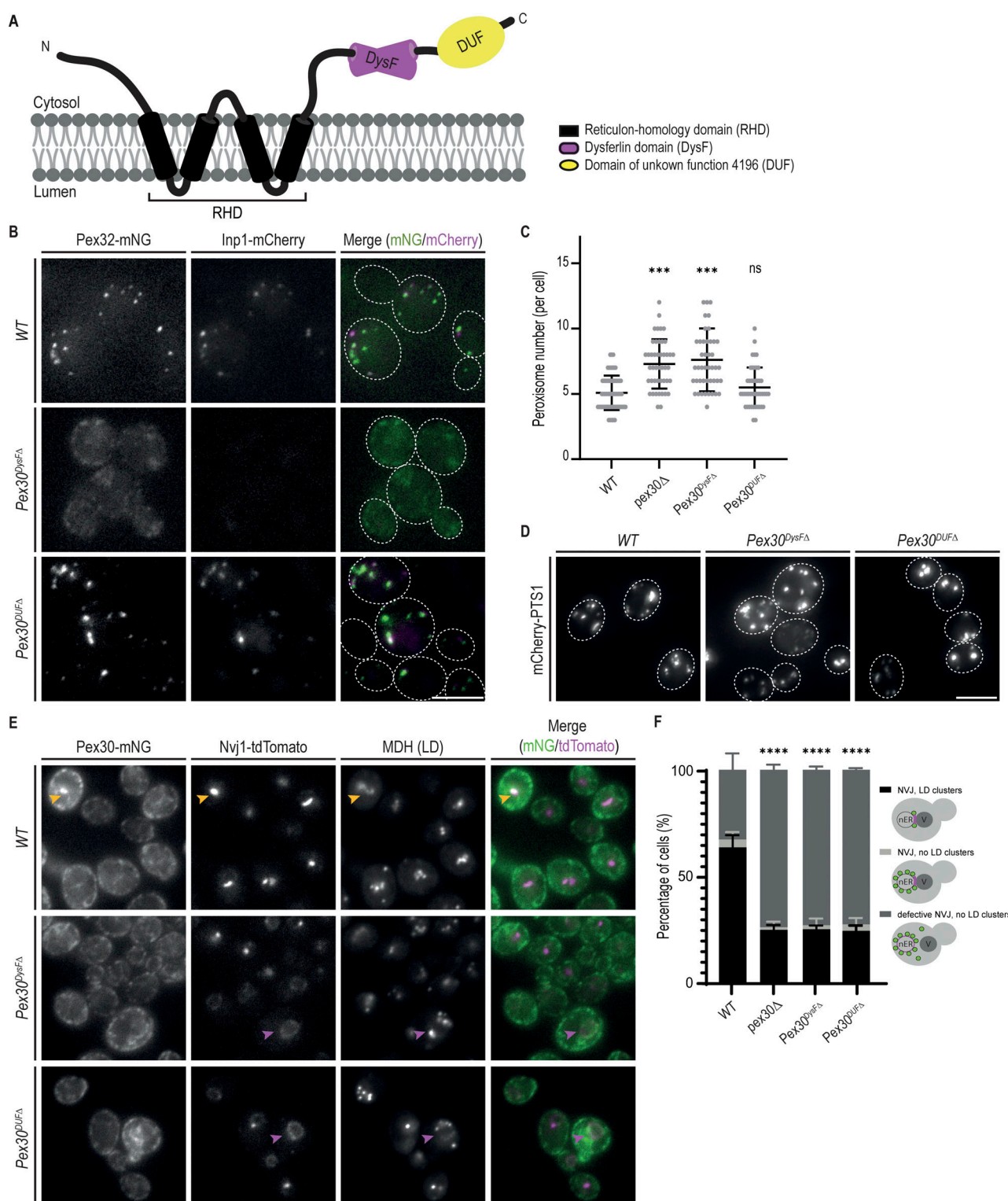

Figure 2. **Distinct requirements of Pex30 DysF and DUF domains at different MCS. (A)** Schematic representation of the predicted Pex30 domains. RHD, reticulon-homology domain; DysF, dysferlin; DUF, domain of unknown function 4196. The ER membrane is shown to indicate Pex30 topology. **(B)** Effect of different Pex30 mutations on the localization of endogenous Pex32-mNG and Inp1-mCherry in exponentially growing cells. Bar, 5 μm. **(C)** Quantification of the number of peroxisomes per cell, in exponentially growing cells. Three independent experiments were analyzed (>30 cells/genotype/experiment were counted). Each dot corresponds to a cell, and the bars represent the mean and SD. Ordinary one-way ANOVA and Dunnett's multiple comparisons were used to compare the number of peroxisomes between mutant and WT cells (***, P < 0.001; ns, not significant). **(D)** Distribution of peroxisomes in cells with the indicated genotype during exponential growth. Peroxisomes were labeled by the mCherry-PTS1 marker. Please note the increase of cytosolic fluorescence in the *Pex30^{DysFΔ}* mutant cells, corresponding to non-imported mCherry-PTS1. Images correspond to maximum intensity Z-projections. Bar, 5 μm. **(E)** Localization of indicated Pex30 mutants tagged with mNG fluorescent protein and expressed from the endogenous Pex30 locus. The localization of Nvj1-tdTomato and LDs,

stained with the neutral lipid dye MDH, was also analyzed in early stationary phase cells. NVJ-clustered and irregularly distributed LDs are indicated by yellow and magenta arrowheads, respectively. Bar, 5 μm. **(F)** Quantification of cells with the indicated genotype displaying defects in NVJ formation and LD clustering during the early stationary phase, as in E. Cells were classified into three categories, as depicted in the cartoons: (1) normal Nvj1 localization and clustered LDs; (2) normal Nvj1 localization and randomly distributed LDs; and (3) abnormal Nvj1 localization and randomly distributed LDs. Three independent experiments were analyzed (>100 cells/genotype/experiment were counted). The bars represent the SD. Ordinary one-way ANOVA and Dunnett's multiple comparisons were used (****, P < 0.0001).

and positively charged (R296, R297, K304) residues, and correspond to a loop between the two antiparallel β-sheets that define the DysF domain. Within this region, the residues W298, I301, K304, F392, and Y395 exhibited the highest frequency of interaction with PA (Fig. 4, B and C).

To test the importance of PA binding in vivo, Pex30 variants with mutations in the putative lipid-binding residues were generated. These Pex30 mutants were expressed at normal levels and did not affect the expression levels of Pex30 adaptors,

indicating that the mutations did not interfere with the assembly of Pex30 complexes (Fig. S3 H). Further analysis of one of these mutants, Pex30^DysF-4A, in which I301, K304, F392, Y395 were mutated to alanine, showed that it localized to the ER like WT Pex30 (Fig. 4 D). However, Pex30^DysF-4A failed to accumulate at the NVJ and resulted in the mislocalization of Nvj1 (Fig. 4 E). This phenotype was reminiscent of that observed in cells expressing Pex30^DysFΔ, which have a complete deletion of the DysF domain. Cells expressing Pex30^DysF-4A also displayed aberrant

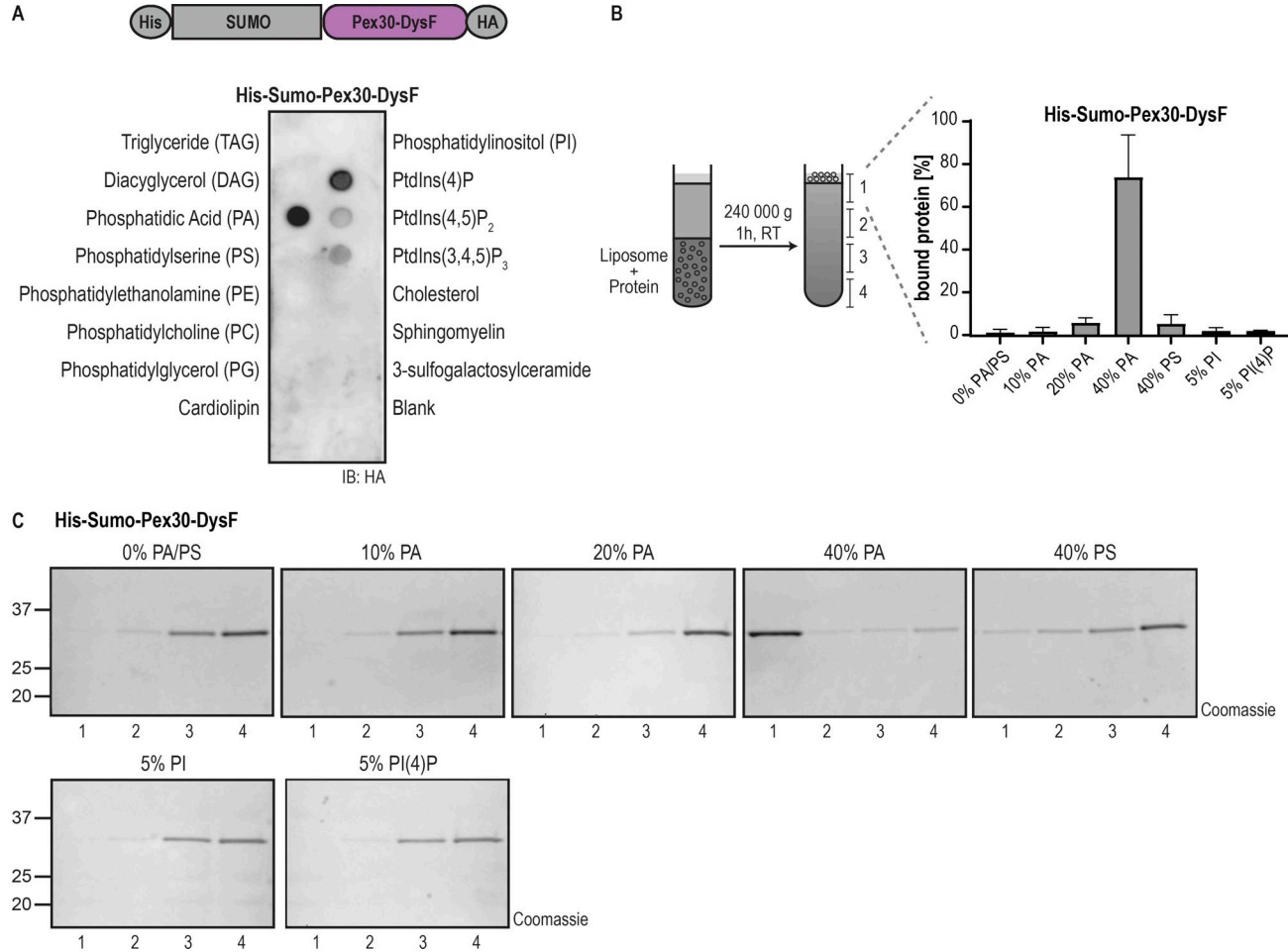

Figure 3.   **DysF is a PA-binding domain. (A)** DysF domain from Pex30 binds to PA and more weakly to phosphoinositides. Purified DysF was incubated with the indicated lipids immobilized in a nitrocellulose membrane. Pex30 DysF was expressed as a fusion protein to the epitope tags indicated and was detected with anti-HA antibody. IB, immunoblot. **(B)** DysF domain from Pex30 binds to PA-containing liposomes. Left: purified DysF and liposomes of defined concentration were pre-mixed for 30 min, layered by sucrose solution, and subjected to centrifugation for liposome flotation, as depicted. The sucrose gradient was fractionated into four fractions, and the samples were subjected to SDS-PAGE and proteins were stained with instant blue. Right: quantification of the percentage of DysF cofractionating with liposomes to the top fraction of experiments shown in C. The bars represent the SD. **(C)** Liposome flotation assay of His-Sumo-Pex30-DysF as described in B using liposomes containing different PA concentrations or the indicated concentration of PS, PI, or PI(4)P. Quantification of the percentage of the protein interacting with the liposomes is represented in B. The position of molecular weight markers (in kDa) is indicated. Source data are available for this figure: SourceData F3.

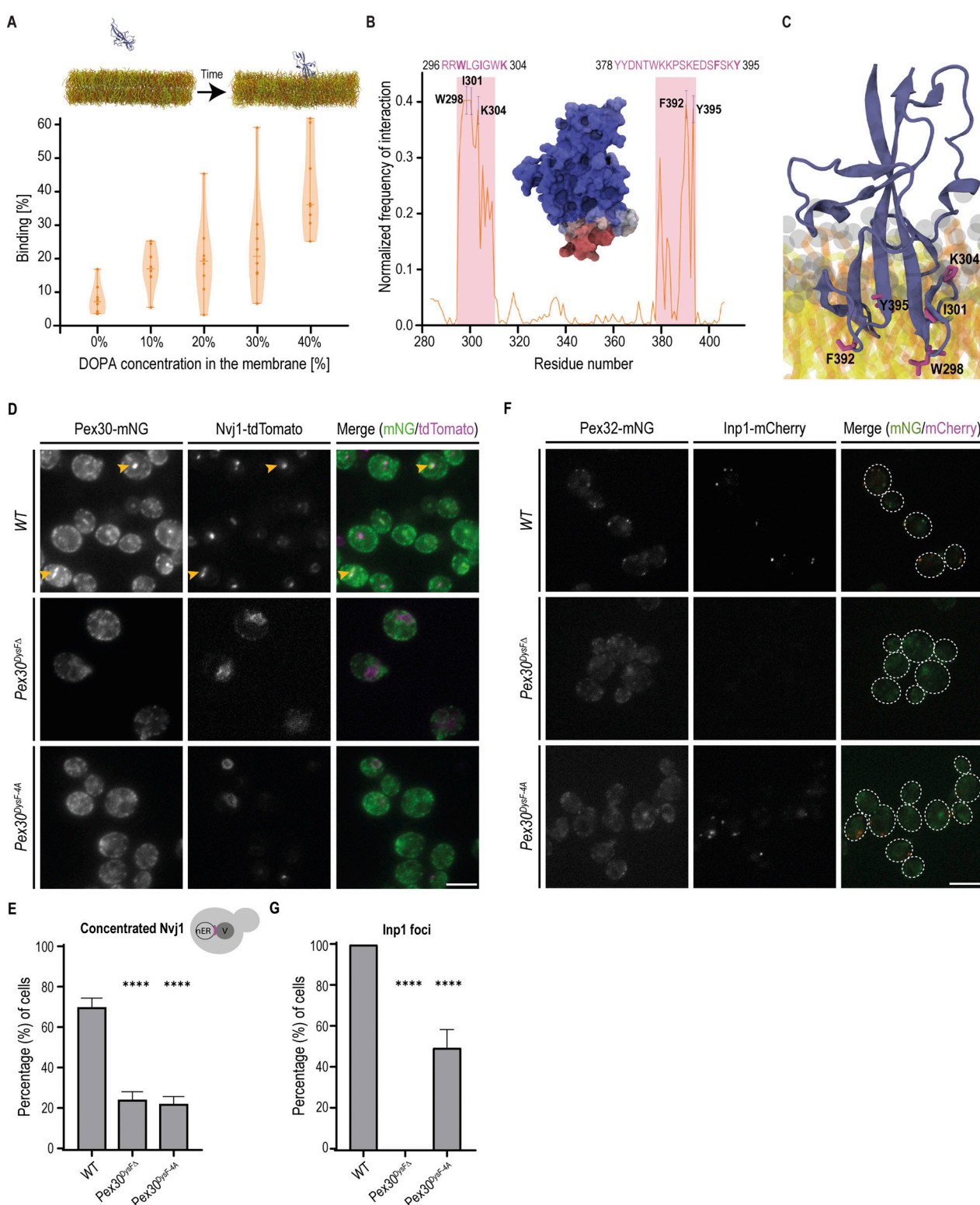

Figure 4. **PA-binding residues of the DysF domain required for Pex30 function. (A)** Binding of the DysF domain of Pex30 to membrane systems with increasing concentrations of DOPA in CG-MD simulations. On top, the visual inset shows the initial system setup and a representative snapshot of the DysF domain bound to the membrane after unbiased CG-MD simulations. **(B)** Normalized frequency of lipid interaction per residue. Representative error bars are shown for the highest binding residues. The binding regions of the protein, with their amino acid sequence shown, are highlighted in pink on the plot. The inset represents the surface of the DysF domain, and residues are colored according to their normalized frequency of interaction with the membrane. **(C)** Representative close-up snapshot of the DysF domain of Pex30 (blue cartoon) bound to a 60% DOPC, 40% DOPA membrane system. Residues displaying high membrane-binding frequency are highlighted in pink. Lipids: gray, PO4 and NC3 beads; yellow, DOPC lipid tails; orange, DOPA lipid tails. **(D)** Localization of the indicated Pex30 mutants tagged with mNG fluorescent protein and of Nvj1-tdTomato in early stationary phase cells. Bar, 5 μm. **(E)** Quantification of cells with

Nvj1-tdTomato concentrated at the NVJ in cells grown as in D. Cells were quantified as depicted in the cartoon by exhibiting organized Nvj1 localization in the nuclear ER. Three independent experiments were analyzed (>100 cells/genotype/experiment were counted). The bars represent the SD. Ordinary one-way ANOVA and Dunnett's multiple comparisons were used (****, P < 0.0001). **(F)** Localization of endogenous Pex32-mNG and Inp1-mCherry in cells expressing the indicated Pex30 mutants. Cells were analyzed in the exponential phase. Bar, 5 μm. **(G)** Quantification of cells with the indicated genotype displaying Inp1-mCherry foci. Three independent experiments were analyzed (>100 cells/genotype/experiment were counted). The bars represent the SD. Ordinary one-way ANOVA and Dunnett's multiple comparisons were used (****, P < 0.0001).

Inp1 localization indicative of defective ER–peroxisome contacts (Fig. 4, F and G). Collectively, these data indicate that DysF is a lipid-binding domain with an affinity for PA and that this activity is critical for Pex30 function.

## Pex30 regulates intracellular PA distribution

The ability of DysF to bind PA suggested a link between Pex30 and PA metabolism (Fig. 5 A). To monitor PA levels in the ER, where Pex30 is localized, we developed Spo20$^{51-91}$-GFP-ER, consisting of the well-established PA biosensor GFP-Spo20$^{51-91}$ (Horchani et al., 2014; Nakanishi et al., 2004) fused to the transmembrane domain of Ubc6, an ER-localized protein (Fig. 5 B). In WT cells, Spo20$^{51-91}$-GFP-ER localized uniformly throughout the ER membrane (Fig. 5, C and D). In *nem1Δ* cells, which have disrupted PA metabolism, Spo20$^{51-91}$-GFP-ER also labeled ER membranes. However, we frequently detected additional foci not observed in WT cells suggesting that Spo20$^{51-91}$-GFP-ER reported on perturbations in PA homeostasis (Fig. 5, C and D). Like *nem1Δ* mutant, *pex30Δ* cells displayed Spo20$^{51-91}$-GFP-ER foci suggestive of abnormal PA metabolism within the ER (Fig. 5, C and D). Consistent with this interpretation, the aberrant distribution of Spo20$^{51-91}$-GFP-ER in *pex30Δ* cells was reversed by the overexpression of Cds1, an enzyme that consumes PA (Fig. 5, A, C, and D). In pex30Δ cells, Spo20$^{51-91}$-GFP-ER foci localized primarily to the cortical ER (Fig. 5 C) and did not coincide with LDs (Fig. S3 I). The distribution of other well-characterized lipid biosensors was similar between *pex30Δ* and *WT* cells, indicating that Pex30 specifically affected the PA probe (Fig. S3, J and K). Moreover, the changes in Spo20$^{51-91}$-GFP-ER distribution in *pex30Δ* cells were not accompanied by complete remodeling of the whole-cell lipidome (Fig. S4 A) (Joshi et al., 2018; Wang et al., 2018) or the lipidome of the vacuole (Fig. S4 B) (Hariri et al., 2019; Manik et al., 2017).

Next, we mutated the DysF domain to test its effect on PA homeostasis in vivo. We observed that deletion of the DysF domain (Pex30$^{DysFΔ}$ cells) or mutation of putative PA-binding residues (Pex30$^{DysF-4A}$ cells) was sufficient to disrupt ER PA homeostasis, as detected by the appearance of Spo20$^{51-91}$-GFP-ER foci (Fig. 5, E and F). Thus, PA homeostasis in the ER depends on Pex30, likely via its DysF domain.

Since the DysF domain is common to all members of the Pex30 family, we asked whether these proteins also contributed to PA homeostasis. Interestingly, mutations in Pex28, Pex29, Pex31, and Pex32 also resulted in the formation of Spo20$^{51-91}$-GFP-ER foci (Fig. 5 G and Fig. S4 C). Thus, the Pex30 protein family appears to have a general role in PA homeostasis, perhaps regulating its distribution.

We wondered whether the role of Pex30 and its family members in PA metabolism was linked to their function in MCS.

To test this hypothesis, we perturbed ER–peroxisome contacts and the NVJ independently of Pex30. Deletion of Inp1 and Pex3, which impair ER–peroxisome MCS, or Nvj1 and Vac8, which are important for the NVJ, resulted in a high fraction of cells with Spo20$^{51-91}$-GFP-ER foci (Fig. 5 H and Fig. S4 D). Interestingly, Spo20$^{51-91}$-GFP-ER foci were not observed in cells lacking Vps13, a bridge-like lipid transfer protein that acts at multiple MCS. Similarly, impairment of other ER functions, such as ergosterol biosynthesis (*hmg1Δ*) and protein quality control (*asi1Δ* and *hrd1Δ*), did not affect Spo20$^{51-91}$-GFP-ER distribution (Fig. 5 H and Fig. S4 E). Together, these data suggest that Pex30-dependent MCS have an important and specific function in ER PA metabolism.

## Phosphorylation regulates Pex30 targeting the NVJ

Pex30 localization to the NVJ is dynamic, with increased accumulation during the diauxic shift (Ferreira and Carvalho, 2021), a period in which cells rewire their metabolism from fermentation to respiration (Kim et al., 2013). To understand the dynamic regulation of Pex30 at the NVJ, we focused on the DUF domain, which is required specifically for the function of Pex30 at this contact site. Several residues within the DUF domain were reported to be phosphorylated, possibly defining a phosphorylation cluster (Fig. S5 A and Table S3). Therefore, we searched for phosphorylation changes that correlated with the accumulation of Pex30 at the NVJ. Mass spectrometry analysis of Pex30 immunoprecipitated from cells in exponential and stationary phases identified many phosphosites, as expected (Fig. S5 B). Among these, only Pex30 serine at position 446 (S446) was significantly increased in stationary phase samples (Fig. S5 B), even when the peptide intensity was normalized to the total Pex30 intensity. To test the role of S446 phosphorylation in Pex30 function, we generated phospho-deficient (Pex30$^{S446A}$) and phospho-mimetic (Pex30$^{S446D}$) mutants in the endogenous Pex30 locus (Fig. 6 A). The resulting proteins were expressed to similar levels to WT Pex30 and interacted normally with Pex29 (Fig. S5 C), the adaptor that targets Pex30 to the NVJ. Consistent with the DUF domain being dispensable for ER–peroxisome MCS, cells expressing Pex30$^{S446A}$ or Pex30$^{S446D}$ had normal peroxisome number and protein import (Fig. S5, D and E).

Next, we tested the effect of S446 phosphorylation on Pex30 accumulation at the NVJ. During exponential growth, the localization of Pex30$^{S446A}$ was indistinguishable from WT Pex30, appearing distributed throughout the ER. However, during the stationary phase, Pex30$^{S446A}$ accumulation at the NVJ was reduced and observed only in a small fraction of cells (Fig. 6 B). Consistent with this defect, NVJ localization of Nvj1, which depends on Pex30 (Ferreira and Carvalho, 2021), was also abnormal (Fig. 6, B and C). Conversely, Pex30$^{S446D}$ accumulated at the

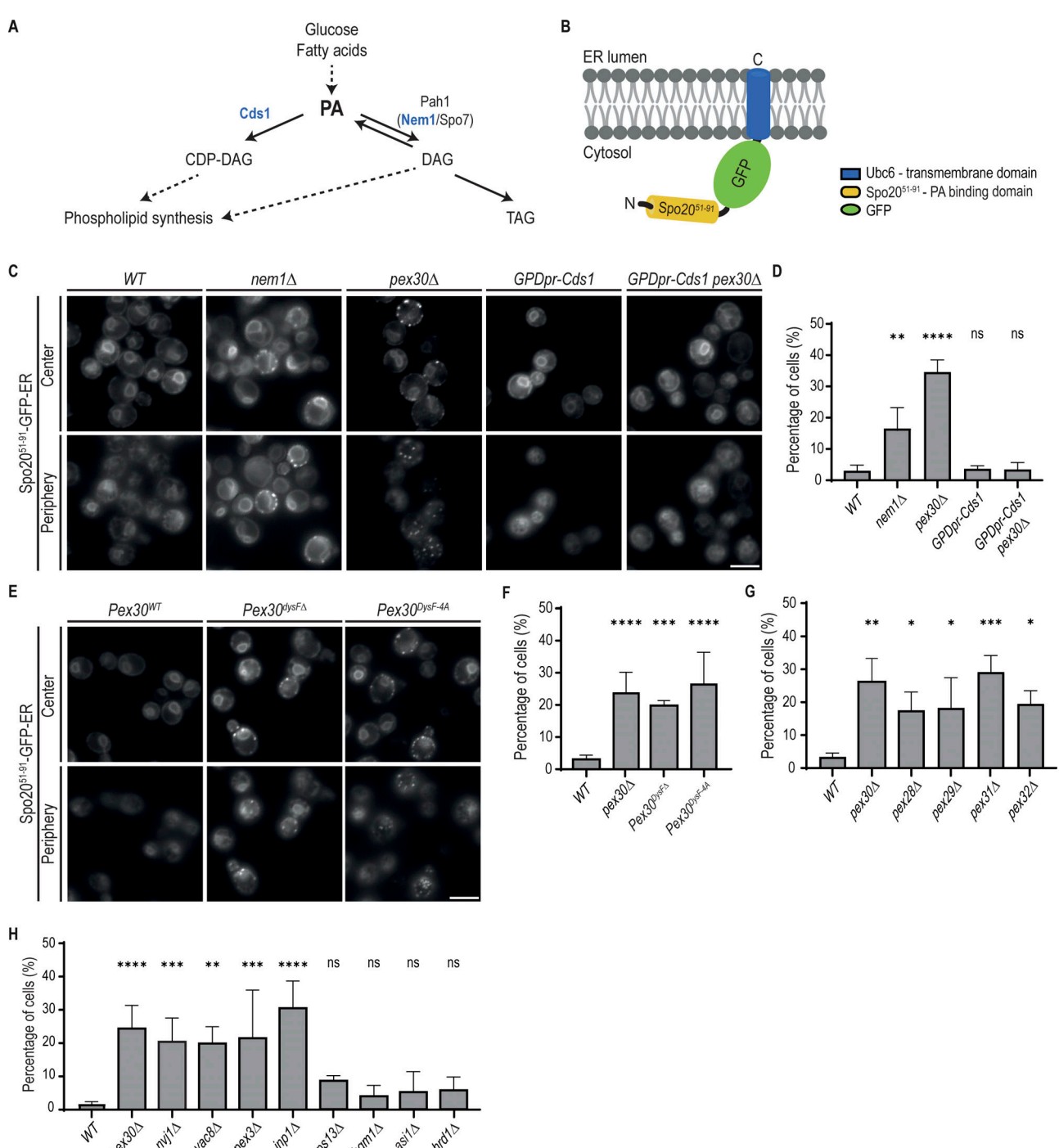

Figure 5. **Pex30 participates in the regulation of the intracellular distribution of PA. (A)** Diagram highlighting the main steps of PA metabolism in yeast. Individual and multiple reactions are indicated by solid and dashed lines, respectively. The enzymes Cds1, Pah1, and the Pah1 activator complex Nem1/Spo7 are indicated. **(B)** Schematic representation of Spo20$^{51–91}$-GFP-ER, an ER-localized PA biosensor. The well-characterized PA sensor Spo20$^{51–91}$-GFP was targeted to the ER by fusing it to the transmembrane domain of Ubc6, an ER-resident protein. **(C)** Localization of Spo20$^{51–91}$-GFP-ER in cells with the indicated genotype. Cells were analyzed during the diauxic shift after overnight growth in SC medium. Individual Z-planes corresponding to the center and the periphery of the cell are shown. Where indicated, CDS1 was expressed from the strong constitutive GPD1 promoter. Bar, 5 µm. **(D)** Quantification of Spo20$^{51–91}$-GFP-ER foci in cells with the indicated genotype grown as in C. The bars represent the mean and SD. Ordinary one-way ANOVA and Dunnett's multiple comparisons were performed to compare the percentage of cells with foci with the WT condition (****, P < 0.0001; **, P < 0.01; ns, not significant). **(E)** Localization of Spo20$^{51–91}$-GFP-ER in cells expressing the indicated Pex30 mutants. Cells were analyzed during the diauxic shift after overnight growth in SC medium. Individual Z-planes corresponding to the center and the periphery of the cell are shown. Bar, 5 µm. **(F)** Quantification of Spo20$^{51–91}$-GFP-ER foci in cells with the indicated genotype grown as in E. The bars represent the mean and SD. Ordinary one-way ANOVA and Dunnett's multiple comparisons were performed to compare the percentage of cells with foci with the WT condition (***, P < 0.001; ****, P < 0.0001). **(G)** Quantification of Spo20$^{51–91}$-GFP-ER foci in cells with mutations in Pex30 family members grown as in E. The bars represent the mean and SD. Ordinary one-way ANOVA and Dunnett's multiple comparisons were performed to compare the percentage of cells with foci with the WT condition (****, P < 0.0001; ***, P < 0.001; **, P < 0.01; *, P < 0.05). **(H)** Quantification of

Spo20[51-91]-GFP-ER foci in cells with the indicated genotype grown as in E. The bars represent the mean and SD. Ordinary one-way ANOVA and Dunnett's multiple comparisons were performed to compare the percentage of cells with foci with the WT condition (****, P < 0.0001; ***, P < 0.001; **, P < 0.01; ns, not significant).

NVJ more efficiently than WT Pex30. In fact, Pex30[S446D] accumulated at the NVJ of a small fraction of exponentially growing cells, a stage where WT Pex30 is never detected at the NVJ. Interestingly, this ectopic NVJ localization of Pex30[S446D] also promoted the localization of Nvj1 (Fig. 6, B and C). The defects in Nvj1 localization appeared specific as the localization of Tsc13, another NVJ component (Kohlwein et al., 2001; Kvam et al., 2005; Tosal-Castano et al., 2021), was unaltered in Pex30 mutants (Fig. S5 F).

As expected, the intracellular distribution of LDs followed the same pattern of localization of Pex30 and Nvj1, confirming that LDs aggregate around the NVJ only when these proteins are properly enriched (Fig. 6 D). Thus, Pex30-S446 phosphorylation is an important determinant in controlling Pex30 localization and NVJ formation. To fully understand the requirements for Pex30 localization to the NVJ, the distribution of Pex30[S446D] and its effect on NVJ formation were analyzed in cells with deletion of DysF or of Pex29. The increased NVJ accumulation of Pex30[S446D] was reversed when this mutation was combined with the deletion of Pex30 DysF or of Pex29 (Fig. 6, E and F). The decrease in Pex30[S446D] accumulation at the NVJ was accompanied by the mislocalization of Nvj1 throughout the nuclear envelope (Fig. 6, E and F). Altogether, these data indicate that multiple independent determinants regulate Pex30 function at the NVJ.

## Discussion

Pex30 is a multidomain protein involved in the organization of ER regions specialized in organelle biogenesis and MCS. The RHD domain, with membrane shaping properties, assembles distinct Pex30 complexes with its MCS adaptors, but how other Pex30 domains contribute to its function remains unknown. Here, we found that the Pex30 DysF domain binds to PA and is important for ER lipid homeostasis, whereas the DUF domain is specifically required for Pex30 function at the NVJ and is regulated by phosphorylation according to nutrient availability. These findings suggest that Pex30 coordinates lipid homeostasis across organelles and in response to cellular metabolism.

Consistent with its localization, deletion of PEX30 resulted in defective ER MCS with both vacuoles and peroxisomes. vEM measurements revealed a reduction in ER–vacuole contacts in cells deleted for Pex30 or its NVJ partner Pex29. A similar defect was observed in cells lacking the tether protein Nvj1, consistent with the previously described role of the Pex30–Pex29 complex in promoting Nvj1 proper localization (Ferreira and Carvalho, 2021). Lack of Pex30 also impaired ER–peroxisome contacts, as assessed by the localization of the contact site protein Inp1. In addition to its localization to ER–peroxisome MCS (Knoblach et al., 2013), Inp1 also localizes to peroxisome–plasma membrane MCS (Hulmes et al., 2020; Krikken et al., 2020). The latter localization depends on interactions of Inp1 N- and C-termini

with plasma membrane phosphoinositides and Pex3 in peroxisomes, respectively, and is critical for the accurate partitioning of peroxisomes during yeast cell division (Hulmes et al., 2020; Krikken et al., 2020). Interestingly, mutations in Pex30 or its ER–peroxisome MCS partner Pex32 caused complete dispersal of Inp1, suggesting that both ER–peroxisome and plasma membrane–peroxisome MCS were disrupted in these mutants. Consistent with this idea, earlier work in *S. cerevisiae* and *H. polymorpha* showed that similar to Inp1 mutations, deletion of Pex32 resulted in defects in peroxisome partition (Knoblach and Rachubinski, 2019; Krikken et al., 2020). These observations suggest a complex interplay between various peroxisome MCS that should be investigated in future studies.

Previously, we observed that *Pex30[DysFΔ]*, a mutant lacking the DysF domain, displays normal protein levels and interactions with the Pex30 partners (Ferreira and Carvalho, 2021). However, *Pex30[DysFΔ]* phenocopied *pex30Δ* cells, which have a complete deletion of Pex30, indicating that DysF was critical for an unknown Pex30 function. In other proteins, DysF domains have been implicated in lipid-related processes (Bansal et al., 2003; Bulankina and Thoms, 2020; Parodi et al., 2015; Nakamura et al., 2017), and in the case of human TECPR1, required for lysosome repair, the DysF domain appears to bind sphingomyelin directly (Boyle et al., 2023; Corkery et al., 2023; Kaur et al., 2023). We now show that the Pex30 DysF domain binds to PA in vitro, a property that is shared with the DysF domains of other Pex30 family members and human Dysferlin. Our in vitro and in silico experiments used soluble versions of DysF domains, requiring high PA concentration for membrane binding. However, in vivo, DysF domains are part of integral membrane proteins, positioned adjacent to the membrane. Moreover, Pex30 family members function as oligomers (Ferreira and Carvalho, 2021), with multiple DysF domains being present within a single Pex30 complex. For these reasons, it is likely that the required PA concentration for the DysF domain to bind to membranes in vivo is lower than the PA concentration we observed in vitro.

PA binding appears to be mediated by a cluster of charged and aromatic residues at one end of the Pex30 DysF domain. Based on our simulations, the bulky aromatic residues may detect and insert into membrane packing defects typical of membranes rich in PA (Vamparys et al., 2013), while the positively charged residues likely contact PA polar heads. Clusters of amino acids with similar properties are present in the DysF domains of other Pex30 family members and are likely the PA-binding activity of these proteins, which also showed ability to bind PA. Interestingly, in TECPR1, the same distal region of the DysF and a similar set of residues are involved in binding to sphingomyelin, a lipid that is not present in yeast. In the future, it will be interesting to dissect the precise determinants of lipid-binding specificity to DysF domains in each case.

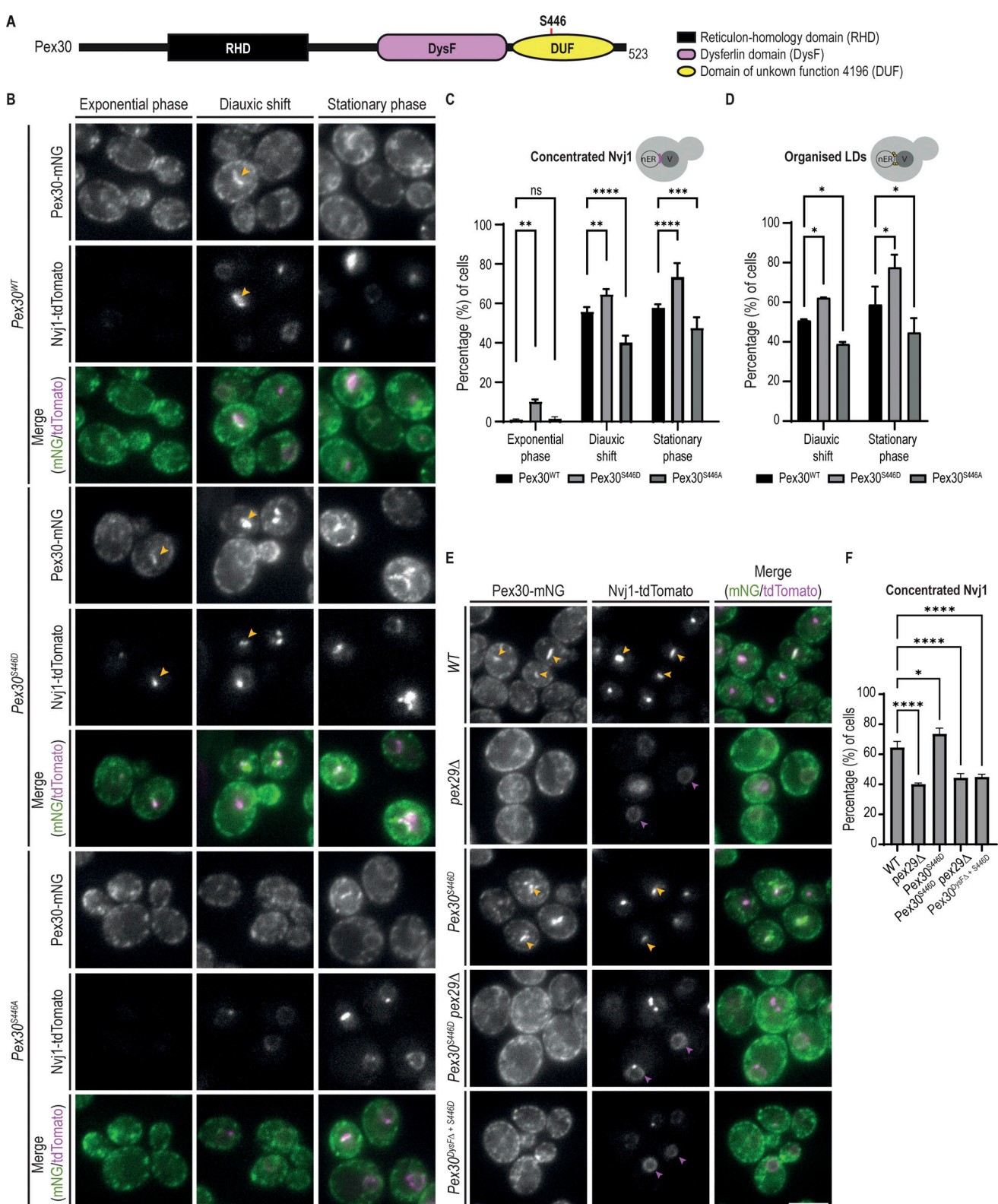

Figure 6. **DUF domain phosphorylation regulates Pex30. (A)** Schematic representation of Pex30 domains. Serine 446 (S446) within the DUF is indicated. **(B)** Localization of Pex30 and the indicated phospho-mutants (Pex30$^{S446D}$ and Pex30$^{S446A}$) expressed from the endogenous Pex30 locus during the exponential, diauxic shift, and stationary phases. Pex30 and derivatives were expressed as mNG fusions. The concentration of endogenous Nvj1-tdTomato at the NVJ was monitored. Yellow arrowheads highlight colocalization between Pex30 and Nvj1. Bar, 5 μm. **(C)** Quantification of cells with Nvj1-tdTomato concentrated at the NVJ, during different growth stages as in B. The bars represent the mean and SD. Ordinary one-way ANOVA and Dunnett's multiple comparisons were performed to compare the percentage of cells with organized Nvj1 with the WT condition for each time point (****, P < 0.0001; ***, P < 0.001; **, P < 0.01; ns, not significant). **(D)** Quantification of cells with LDs clustered around the NVJ, during the diauxic shift and stationary phase. The bars represent

the mean and SD. Ordinary one-way ANOVA and Dunnett's multiple comparisons were performed to compare the percentage of cells with clustered LDs with the WT condition for each time point (*, P < 0.05). **(E)** Localization of Pex30 and Nvj1-tdTomato in cells with the indicated genotype during the diauxic shift. Yellow arrowheads highlight colocalization between Pex30 and Nvj1. Magenta arrowheads highlight mislocalized Nvj1-tdTomato. Bar, 5 μm. **(F)** Quantification of cells with Nvj1-tdTomato concentrated at the NVJ, in cells grown as in E. The bars represent the mean and SD. Ordinary one-way ANOVA and Dunnett's multiple comparisons were performed to compare the percentage of cells with organized Nvj1 with the WT condition (****, P < 0.0001; *, P < 0.05).

Whole-cell lipid analysis has failed to detect major changes in the lipidome of *pex30Δ* cells (Fig. S4 A) (Joshi et al., 2018; Wang et al., 2018). However, these cells showed aberrant distribution of an ER PA reporter, which appeared in small foci dispersed throughout the cortical ER regions. The effect was specific, and the distribution of a variety of other lipid biosensors appeared unaltered. Similar defects in the distribution of the PA biosensor were observed in *Pex30^{DysFΔ}* and *Pex30^{DysF-4A}* cells, directly linking the PA defects to the Pex30 DysF domain. Mutations in all Pex30 family members resulted in similar defects in PA metabolism, suggesting that this protein family may have a general role in PA homeostasis.

More work will be needed to dissect the precise nature of the defect, but the PA distribution phenotype in *pex30Δ* mutants appears to be related to its function at ER MCS. Mutations in components of the NVJ or ER contacts with peroxisomes linked to Pex30 function resulted in similar defects in PA metabolism. In contrast, no effect was observed in cells lacking Vps13, which is critical for bulk lipid transport between organelles. Whether Pex30 and other family members facilitate PA transfer at MCS should be tested in future studies. While they appear to lack an obvious hydrophobic cavity normally observed in lipid transfer proteins, Pex30 family proteins may stimulate the process by some other mechanisms.

The changes in PA metabolism observed in *pex30Δ* mutants may also impact the membrane shape. PA is a non-bilayer phospholipid and tends to accumulate in regions of high membrane curvature. Considering the importance of Spo7 in PA regulation, the suppression of *rtn1Δrtn2Δyop1Δspo7Δ* lethality by Pex30 overexpression may be related to its function in PA metabolism via its DysF domain, besides its role in membrane shaping via the RHD as initially proposed (Joshi et al., 2016). Other links between PA metabolism and Pex30 came from studies on LDs, particularly with mutations in the seipin protein Sei1, a critical LD assembly factor (Joshi et al., 2018; Wang et al., 2018). Similar to *pex30Δ* cells, *sei1Δ* mutants display abnormal distribution of PA biosensors, with Pex30 aberrantly localized to PA-rich membranes (Grippa et al., 2015; Wolinski et al., 2015). Importantly, Pex30 relocalization, which requires its DysF domain (Ferreira and Carvalho, 2021), is functionally important since simultaneous deletion of Pex30 and Sei1 strongly affects ER morphology and compromises cell viability (Wang et al., 2018). While these data implicate Pex30 in both PA homeostasis and ER morphology, the mechanistic details by which Pex30 functions to regulate organelle organization and communication require further investigation.

We showed that Pex30 is regulated by phosphorylation, a modification that affected a pool of Pex30 specifically at the NVJ. This modification in the DUF domain, present in Pex30 but not in other family members, occurs in response to changes in nutrient availability, in line with other observations linking the dynamics of MCS to cellular metabolic rewiring (Bohnert, 2020; Klemm, 2021; Voeltz et al., 2024). The kinase involved in S446 modification remains unknown and should be investigated in the future. Pex30 localization to the NVJ upon S446 phosphorylation requires a functional DysF suggesting the involvement of PA in the recruitment of Pex30. The importance of PA homeostasis at the NVJ during the diauxic shift is underscored by the recruitment of Pah1, the main PA hydrolase, to these MCS (Barbosa et al., 2019). Pah1 is also regulated by phosphorylation, but whether this controls its localization to the NVJ is unknown (Su et al., 2014). However, both mutations in Pex30 or Pah1 result in defects in NVJ formation and LD organization (Barbosa et al., 2019; Ferreira and Carvalho, 2021). Other nutritional stresses, such as acute glucose depletion, also induce remodeling of the NVJ with the accumulation of certain sterol enzymes such as Hmg1 and Hmg2. This process also requires Nvj1, but whether Pex30 and Pah1 are involved remains unknown. Several studies including ours identified additional other potential phosphosites in Pex30 (Table S3). While S446 appears to control specifically Pex30 at the NVJ, other sites are likely to control other functions of Pex30, such as regulation of ER–peroxisome MCS, and should be further characterized in the future.

## Materials and methods

### Antibodies

Pex30 antibody (1:1,000; rabbit polyclonal) was raised against the C-terminal peptide TEEKEQSNPTIGRDS (Eurogentec). mNG antibody (dilution 1:1,000; rabbit polyclonal 53061S) was purchased from Cell Signaling. RFP/mCherry antibody (dilution 1: 1,000; rabbit polyclonal ab62341) was purchased from Abcam. Dpm1 antibody (dilution 1:10,000; mouse monoclonal 5C5A7) was purchased from Invitrogen. Myc antibody (dilution 1:1,000; mouse monoclonal 9E10) was purchased from Roche. HA antibody (dilution 1:2,000; rat monoclonal 3F10) was purchased from Roche. V5 antibody (dilution 1:5,000; rabbit monoclonal D3H8Q) was purchased from Cell Signaling.

### Yeast strains and plasmids

Yeast strains used in this study are isogenic either to BY4741 (*MATa ura3Δ0 his3Δ1 leu2Δ0 met15Δ0*) or to BY4742 (*MATα ura3Δ0 his3Δ1 leu2Δ0 lys2Δ0*) and are listed in Table S1. Tagging of proteins, replacement of promoter, and individual gene deletions were performed by standard PCR-based homologous recombination (Longtine et al., 1998; Janke et al., 2004). Point mutations, protein tagging, replacement of regions within the ORF for partial gene deletions, and full gene deletions were performed by CRISPR-based gene editing (adapted from Laughery et al. [2015]). Briefly, a single guide RNA (sgRNA) sequence

targeting the desired region of the gene of interest was designed using the online software Benchling [Biology Software] (2021) and http://wyrickbioinfo2.smb.wsu.edu. The sgRNA was cloned into the pML107 vector, containing a Cas9 endonuclease from *Streptococcus pyogenes*. This plasmid along with a PCR-amplified template containing the desired modification was transformed using a standard yeast transformation protocol. Strains with multiple deletions/tags were obtained by crossing haploid cells of opposite mating types, followed by sporulation and tetrad dissection using standard protocols (Guthrie and Fink, 1991). The yeast strains and plasmids used in this study are listed in Tables S1 and S2, respectively.

## Growth conditions

Cells were grown at 30°C in YPD liquid medium (1% Bacto yeast extract, 2% Bacto peptone, 2% glucose) or synthetic complete medium (0.67% yeast nitrogen base with ammonium sulfate, 0.06% Complete Supplement Mixture without histidine, leucine, tryptophan, and uracil, 2% glucose) supplemented with required amino acids (300 µM histidine, 1,680 µM leucine, 400 µM tryptophan, 200 µM uracil), unless indicated otherwise. For microscopy, protein analysis, and immunoprecipitation experiments, exponentially growing cells were analyzed/collected at an $OD_{600}$ of 1; cells grown to the diauxic shift or early stationary phase were analyzed/collected about 7–8 h after an $OD_{600}$ of 1; and cells in the stationary phase were analyzed/collected 24 h after an $OD_{600}$ of 1.

## Fluorescence microscopy

Wide-field epifluorescence microscopy was performed at room temperature using the Zeiss Axio Observer.Z1 equipped with a digital CMOS camera (ORCA Flash 4.0; Hamamatsu), controlled by 3i Slidebook 6.0 software. A Plan-APOCHROMAT 63× 1.4 objective or a Plan-APOCHROMAT 100× 1.4 objective was used with immersion oil, and stacks of images spaced 0.11 µm (11 slides) were acquired, but only one Z-plane was shown. When Z-projection was performed, it is mentioned in the figure legend. The images were normalized using 3i Slidebook 6.0 software and exported as TIFF files.

BODIPY, GFP, and mNG signals were detected using a GFP fluorescence setup consisting of a 485/20-nm band-pass excitation filter (Zeiss) and a 525/30-nm band-pass emission filter. Monodansyl pentane (MDH) was detected using a DAPI fluorescence setup consisting of a 385/1-nm band-pass excitation filter (Zeiss) and a 440/40-nm band-pass emission filter. The tdTomato and mCherry signals were detected using a mCherry fluorescence setup consisting of a 560/25-nm band-pass excitation filter (Zeiss) and a 607/36-nm band-pass emission filter. All setups include a 410/504/582/669-Di01 quad dichroic mirror. LDs were stained with the neutral lipid dyes BODIPY 493/503 (Invitrogen) and MDH (Abgent), by being incubated for 10 min (1 µg/ml BODIPY; 0.1 mM MDH) at room temperature, pelleted at 5,000 *g* for 3 min, and resuspended in synthetic media.

## vEM

### Sample preparation for vEM

Electron microscopy was performed on spheroplasts in order to improve staining efficiency and resin infiltration of cells in the stationary phase. 50 $OD_{600}$ of cells were washed in water and resuspended in 5 ml of resuspension buffer (0.1 M Tris-HCl, pH 9.5, 10 mM DTT). After 10 min shaking at 25°C, cells were washed in spheroplasting buffer (0.7 M sorbitol, 0.5% glucose, 10 mM Tris-HCl, pH 7.4, 1% yeast extract, 2% Bacto peptone, 1 mM phenylmethylsulfonyl fluoride [PMSF]). Cells were resuspended in 1 ml of spheroplasting buffer and Zymolyase 20T (10 µg/OD) to digest the cell wall and incubated at 30°C for 30 min. An equal volume of 2x strength fixative was added to cells for a final concentration of 2% glutaraldehyde + 1% formaldehyde + 0.2 M sorbitol, 2 mM $MgCl_2$, 2 mM $CaCl_2$ in PIPES buffer, pH 6.8. Samples were fixed for 5 min at room temperature, then gently spun down, and resuspended in 2 ml of fixative. Samples were fixed for 2 h at room temperature and then stored at 4°C overnight. All subsequent steps were performed at room temperature with rotation unless otherwise noted. Samples were washed twice in 0.1 M PIPES buffer, pH 6.8, for 10 min and pelleted at 7,000 rpm for 30 s each time. Cells were then resuspended in warm 2.5% low melting point agarose, pelleted at 4,000 rpm for 20 s, then 7,000 rpm for 30 s, and incubated in the fridge for 15 min. The agarose was then cut into 1-mm³ pieces for ease of processing and resuspended in 0.1 M PIPES buffer containing 50 mM glycine to quench free aldehydes and minimize the formation of osmium precipitates. Samples were washed in 0.1 M PIPES buffer for a further 5 min and then stained with 2% osmium tetroxide + 1.5% potassium ferrocyanide in 0.1 M PIPES buffer for 1 h at 4°C. Samples were then washed three times for 10 min with Milli-Q water, stained in 1% thiocarbohydrazide for 20 min, washed as before, and then stained with 2% osmium tetroxide for 30 min. Following a further six times of washing for 10 min in water, the samples were incubated overnight in 1% uranyl acetate aq. at 4°C, washed three times for 10 min with water, stained with warm lead aspartate for 60 min, and then washed again with water three times for 10 min. Samples were then sequentially dehydrated with ice-cold 30%, 50%, 70%, and 90% ethanol at 4°C with rotation for 10–20 min each, then twice with 100% ethanol for 20 min each followed by two 10-min incubations in ice-cold pure acetone. Resin infiltration was performed in acetone at 4°C with rotation using Quetol 651 resin of the following formula: 15g Quetol 651 monomer, 25g nonenyl succinic anhydride, 4g nadic methyl anhydride, and 1g DMP-30. Samples were infiltrated with 25% resin for 1.25 h, then 50% resin overnight. Samples were then spun at 10,000 rpm for 30 min at 4°C, then incubated in 75% resin for 4 h, spun as before, and then incubated in pure resin overnight. Three more incubations in pure resin were performed for 8–12 h each at room temperature with centrifugation at 10,000 rpm for 30 min after each step. Agarose pieces were then embedded in Beem capsules filled with fresh resin and polymerized for 72 h at 60°C.

### Serial block-face sectioning scanning electron microscopy

Resin blocks were trimmed down and mounted onto a 3View pin using silver conductive epoxy, then sputter-coated with ~15 nm of gold. Serial block-face sectioning scanning electron microscopy was performed using a Zeiss Merlin Compact equipped with a Gatan 3View system and a Gatan OnPoint backscatter

detector. The microscope was operated at 1.8 kV with a 30-μm aperture and the Focal Charge Compensation device at 100%. Images were acquired at a resolution of 3 × 3 × 100 nm (x,y,z) using a 3-μs pixel dwell and a frame size of 8,000 × 8,000 pixels.

Randomly selected cells were analyzed using all the respective Z-slices covering the entirety of the cell volume. Multiple measures of the distance between the nucleus and the vacuole were done using Fiji/ImageJ (Schindelin et al., 2012), but only the shortest distance was used in the quantifications. Similarly, the number of vacuoles per cell was manually quantified using all the Z-slices.

### Focused ion beam scanning electron microscopy
Pins were prepared as described above. Focused ion beam scanning electron microscopy was performed using a JEOL 4700F. The focused ion beam was set to 30 kV with a probe current of 8, and the electron beam was set to 3 kV with a probe current of 10. Images were acquired using the backscattered detector at a scan speed of 3 with four line integration averages, a pixel size of 9 × 9 × 20 nm (x,y,z), and a frame size of 1,280 × 960 pixels. The resulting image stack was cropped and aligned using an affine transformation in Fiji (Schindelin et al., 2012), where the contrast was also normalized and LUT inverted.

In our samples, nuclei, vacuoles, and LDs were large and contrast-rich organelles, which allowed fast manual segmentation using IMOD (Kremer et al., 1996). The manual segmentation was performed using interpolation every six slices. 3D rendering was done also with IMOD. Videos of morphology and 3D rendering were exported using Fiji at five frames per second.

### Protein analysis
For western blotting, whole-cell extracts of exponentially growing cells or cells grown to the diauxic shift were prepared from 2 OD units of cells. Pelleted cells were resuspended in 300 μl of 0.15 M NaOH and incubated on ice for 10 min. After centrifugation at a maximum speed for 2 min at 4°C, the pellet was resuspended in sample buffer and heated at 65°C for 10 min. Proteins were separated by SDS-PAGE in Criterion TGX precast gels (Bio-Rad), transferred to a polyvinylidene difluoride membrane, and analyzed with the indicated antibodies. Proteins were detected using Western Lightning Pro chemiluminescent substrate (Revvity). Protein levels on western blots were quantified using ImageStudio 5.5 (LI-COR).

### Immunoprecipitation
Immunoprecipitation of endogenously tagged Pex28-13xMyc, Pex29-V5, and Pex32-3xHA was performed as follows. Approximately 100 $OD_{600}$ units of yeast culture grown in YPD were harvested by centrifugation at 3,900 × $g$, washed, and resuspended in 700 μl of lysis buffer (LB; 50 mM Tris-HCl [pH 7.4], 200 mM NaCl, 1 mM EDTA, 1 mM PMSF, and cOmplete protease inhibitor [Roche]). Cells were lysed with glass beads, and lysates were cleared by low-speed centrifugation at 4°C. Membranes were pelleted at 45,000 $g$ for 25 min at 4°C in an Optima Max tabletop ultracentrifuge on a TLA-100.3 rotor (Beckman Coulter). The crude membrane fraction was resuspended in 600 μl LB. Then, 700 μl of LB supplemented with decyl maltose neopentyl glycol (DMNG) was added to obtain 1%

final concentration, and membranes were solubilized for 2 h on a rotating wheel at 4°C. Solubilized membranes were cleared for 15 min at 4°C at full speed in a tabletop centrifuge. The tagged proteins were affinity-isolated by incubation for 2 h at 4°C with anti-HA magnetic beads (Thermo Fisher Scientific), anti-V5 magnetic beads, and anti-Myc magnetic trap (ChromoTek). Beads were washed three times with 0.02% DMNG in LB, eluted with Laemmli buffer, and analyzed by SDS-PAGE and immunoblotting. In all experiments, the input lane corresponds to 10% of the total extract used for immunoprecipitation.

### Protein analysis by label-free quantitative mass spectrometry
#### Sample processing protocol
Yeast cultures diluted from a starter culture were grown overnight in YPD, and samples in the exponential phase were collected at $OD_{600} = 1$, while samples in the stationary phase were collected ~7.5–8 h after $OD_{600} = 1$ ($OD_{600} = 4$–5). 200 $OD_{600}$ units of cells were pelleted and washed in cold PBS. Cell lysates were prepared in LB (50 mM Tris-HCl, pH 7, 200 mM NaCl, 1 mM EDTA, 1 mM PMSF), supplemented with cOmplete protease inhibitor (Roche) and the phosphatase inhibitor PhosSTOP (Roche), and lysed with glass beads in bead mill homogenizer with 6 cycles of 30 s, speed 4.0 m/s, and 60-s breaks, and the lysates were cleared by low-speed centrifugation at 4°C, 15 min. The lysate was supplemented with an equivalent volume of LB supplemented with DMNG to obtain 1% final concentration, and membranes were solubilized for 2 h on a rotating wheel at 4°C. Solubilized membranes were cleared for 15 min at 4°C at full speed in a tabletop centrifuge. The tagged proteins were affinity-isolated by incubation for 2 h at 4°C with 60 μl of previously equilibrated anti-mNG magnetic agarose (ChromoTek). The agarose was washed five times with 0.02% DMNG in LB. 10% of the beads were removed and eluted with Laemmli buffer, to be analyzed by SDS-PAGE and immunoblotting.

Protein samples were prepared using the suspension trapping (S-Trap) sample preparation method (Zougman et al., 2014). Briefly, beads were suspended in SDS-LB (50 mM triethylammonium bicarbonate [TEAB], pH 7.55, 50 mg/ml SDS) and sonicated for 1 min in a water bath sonicator, and beads were removed using a magnet. Disulfide bonds in proteins were reduced using 20 mM TCEP for 15 min at 47°C, and cysteines were alkylated for 15 min at room temperature, in the dark, using 20 mM chloroacetamide. The protein was precipitated by the addition of 12% phosphoric acid, followed by the S-Trap binding buffer (100 mM TEAB, 90% methanol). The sample was loaded onto S-Trap micro spin columns (C02-micro, ProtiFi) and centrifuged at 4,000 $g$ for 1 min. The spin column was washed five times with S-Trap binding buffer, and the S-Trap column was subsequently moved to protein low-bind tubes (Eppendorf). Lysine-C (Promega) and trypsin (Promega) in a 1:50 protein ratio in 50 mM TEAB, pH 8, were added to each S-Trap micro spin column to digest the proteins for 3 h at 47°C. Peptides were eluted with 40 μl of 50 mM TEAB, followed by 40 μl of 0.2% formic acid and 40 μl of 50% acetonitrile in 0.2% formic acid, by centrifugation at 4,000 $g$ for 1 min. Samples were dried in a speed vacuum and stored at −20°C until mass spectrometry analysis.

## Sample analysis by mass spectrometry

Peptides were resolubilized in 9 μl buffer A (0.1% formic acid in mass spectrometry grade water) and 1 μl buffer A* (0.1% formic acid, 0.1% TFA in MS grade water). Peptides were separated using an Easy-nLC 1200 system (Thermo Fisher Scientific) coupled with a Q Exactive HF mass spectrometer via a Nanospray Flex ion source. The analytical column (50 cm, 75 μm inner diameter [New Objective] packed in-house with C18 resin ReproSil-Pur 120, 1.9 μm diameter [Dr. Maisch]) was operated at a constant flow rate of 250 nl/min. Gradients of 90 min were used to elute peptides (Solvent A: aqueous 0.1% formic acid; Solvent B: 80% acetonitrile, 0.1% formic acid). MS1 spectra with a mass range of 300–1,650 m/z were acquired in profile mode using a resolution of 60,000 (maximum fill time of 20 ms or a maximum of 3e6 ions [automatic gain control, AGC]). Fragmentation was triggered for the top 15 ions with charge 2–8 on the MS scan (data-dependent acquisition) with a 30-s dynamic exclusion window (normalized collision energy was 28). Precursors were isolated with a 1.4-m/z window, and MS/MS spectra were acquired in profile mode with a resolution of 15,000 (maximum fill time of 80 ms, AGC target of 2e4 ions).

## Data processing

All mass spectra were analyzed using MaxQuant 1.6.3.4 (Cox et al., 2011; Cox and Mann, 2008) and searched against the *S. cerevisiae* reference proteome (UP000002311) downloaded from UniProt. Peak list generation was performed within MaxQuant, and searches were performed using default parameters and the built-in Andromeda search engine (Cox et al., 2011). The enzyme specificity was set to consider fully tryptic peptides, and two missed cleavages were allowed. Oxidation of methionine, N-terminal acetylation, and phospho (STY) were set as variable modifications. Carbamidomethylation of cysteine was set as a fixed modification. A protein and peptide false discovery rate of <1% was employed in MaxQuant. Reverse hits and contaminants were removed before downstream analysis. Phosphopeptides were filtered for localization probability >0.75 and presence in at least two of three replicates. Pex30 phosphopeptide intensity was normalized based on total Pex30 protein intensity. Statistical significance was determined using Student's *t* test (two-tailed).

## Purification of soluble lipid-binding domains

Sequences of various DysF domains and Opi1-Q2 were codon-optimized for bacterial expression and inserted into a vector containing an N-terminal His-SUMO tag. This plasmid was transformed into BL21-CodonPlus (DE3)-RIPL Competent *E. coli* (Agilent). A preculture was grown in 50 ml of LB supplemented with 50 μg/ml kanamycin (Merck) and 50 μg/ml chloramphenicol (Sigma-Aldrich) at 37°C, 200 rpm. Overnight cultures were diluted into 1 liter of terrific broth (24 g/liter yeast extract, 20 g/liter tryptone, 4 ml/liter glycerol), supplemented with kanamycin and K-Phos salt solution (0.17M KH2PO4, 0.72M K2HPO4). Expression was induced at $OD_{600}$ 0.5–0.6 by adding 0.5 mM IPTG at 30°C for 3 h. Cell pellets were obtained by centrifugation at 4,000 *g* for 10 min, followed by resuspension

in LB (50 mM Tris-HCl, pH 8, 500 mM NaCl, 30 mM imidazole, pH 8.0, 1 mM PMSF). Cells were enzymatically digested with lysozyme (Sigma-Aldrich) and DNase I (Roche) for 30 min, followed by sonication. Non-lysed cells were removed by a low-speed spin at 4,000 *g* for 10 min. The supernatant was subjected to ultracentrifugation in a Ti45 rotor, at 40,000 rpm for 45 min, at 4°C. The supernatant was incubated with 3 ml of Ni-NTA beads (Thermo Fisher Scientific) overnight at 4°C with slow rotation. The beads were washed with 20-column volume of LB, and the protein was eluted with elution buffer (20 mM Tris-HCl, pH 8, 200 mM NaCl, 10 mM imidazole) by adding 1 ml each time to a total of 10 ml. One of the elution fractions was run with AKTA Pure (GE Healthcare) on a 24-ml Superdex 200 10/300 size-exclusion column (Cytiva) in AKTA buffer, at 0.5 ml/min, collecting 1 ml aliquots. The His-Sumo peptide was obtained by on-bead digestion with in-house–purified Ulp1.

## Protein–lipid overlap assay

The protein–lipid overlap assay was performed as adapted from previously described methods (Dowler et al., 2002). Membrane strips prespotted with lipids were purchased from Echelon Biosciences. After half an hour at room temperature, the membranes were first blocked with 5% milk in PBS-T (1% Tween-20) for 1 h and then incubated with 1 μg/ml of purified protein domain in the blocking buffer at 4°C overnight. The blots were washed three times with PBS-T and incubated in 1% milk in PBS-T with an antibody against the C-terminal tag of the purified protein for 2 h at room temperature. The membranes were washed three times with PBS-T and soaked with 1% milk in PBS-T with the secondary antibody for 45 min at room temperature. After washing with PBS-T, the protein was detected using Western Lightning Pro chemiluminescent substrate (Revvity).

## Liposome preparation

Synthetic lipids 1-palmitoyl-2-oleoyl-glycero-3-phosphocholine (16:0-18:1 POPC, 850457; Avanti Polar Lipids), 1-palmitoyl-2-oleoyl-sn-glycero-3-phosphate (16:0-18:1 POPA, 840857; Avanti Polar Lipids), 1-palmitoyl-2-oleoyl-sn-glycero-3-phospho-L-serine (16:0-18:1 POPS, 840034; Avanti Polar Lipids), 1,2-dioleoyl-sn-glycero-3-phospho-(1′-myo-inositol) (18:1 PI, 850149; Avanti Polar Lipids), and 1,2-dioleoyl-sn-glycero-3-phospho-(1′-myo-inositol-4′-phosphate) (18:1 PI(4)P, 850151; Avanti Polar Lipids) were obtained commercially. Lipid stocks of each lipid were prepared in chloroform to yield 25 mg/ml. The desired POPC:PA/PS lipid ratios were mixed in a small glass flask and dried under argon stream, forming lipid films that were stored at –20°C. On the day of the experiment, the lipid films were rehydrated for 2 h at room temperature using NaPi buffer (20 mM sodium phosphate, 150 mM NaCl, pH 7.4) to a final lipid concentration of 4 mM and mixed every 15 min. After rehydration, the liposomes were subjected to five freeze/thaw cycles using liquid nitrogen and a 50°C heating block, followed by stepwise extrusion of the liposomes using the Mini-extruder (Avanti Polar Lipids) for 21 passages through a 100-nm pore size filter (Avanti Polar Lipids) to produce monodisperse populations of mostly unilamellar vesicles.

## Liposome flotation assay

Liposomes and purified protein were mixed in a molar protein: lipid ratio of 1:3,300 in a total volume of 150 µl and incubated for 30 min at room temperature in an ultracentrifugation tube (Beckman Coulter) (adapted from Hofbauer et al. [2018]). After incubation, 100 µl of 75% sucrose (Sigma-Aldrich) dissolved in LF buffer (25 mM HEPES, 10 mM sodium phosphate, 150 mM sodium chloride, and 0.5 mM EDTA, pH 7.4) was added and gently mixed with the sample to produce a final concentration of 30% sucrose. 200 µl of 20% sucrose dissolved in LF buffer was carefully layered on top of the 30% sucrose fraction, and subsequently, 50 µl LF buffer was layered on top of the 20% sucrose fraction, resulting in 500 µl of total volume. Sucrose density gradient centrifugation was conducted for 1.5 h at 22°C at a speed of 75,000 rpm in an Optima MAX-XP ultracentrifuge using a TLA 100.3 rotor (Beckman Coulter). Four fractions of 125 µl were collected from the top of the tube. Each fraction was mixed with 3× reducing protein sample buffer and boiled for 15 min at 65°C, and then, 15 µl of each fraction was subjected to SDS-PAGE using gradient gels (4–20%; Bio-Rad). The gels were stained with InstantBlue Coomassie Protein Stain (Abcam) to visualize proteins and lipids. Protein bands were quantified using ImageStudio 5.5 (LI-COR). The bound fraction was determined as the amount of protein in the top fraction divided by the total protein content in all fractions together.

## Mass spectrometry lipidomics
### Vacuole isolation

For lipidomics experiments, a yeast strain was generated with a bait tag targeted to the C terminus of Mam3, a vacuolar membrane protein involved in magnesium sequestration (Tang et al., 2022) and deletion mutants of PEX30 or PEX29 from this strain were also produced. The bait tag for immunoisolation contains a linker region followed by a Myc epitope tag for detection in immunoblotting analysis, a specific cleavage site for the human rhinovirus 3C (HRV-3C) protease for selective elution from the affinity matrix, and three repeats of a FLAG epitope that ensures binding to the affinity matrix. From a starter culture in the exponential phase, 5 liters of synthetic complete medium was inoculated to grow overnight, at 30°C and 200 rpm, until $OD_{600}$ = 1. Cells were collected in the stationary phase, exactly 24 h after that time point. The final $OD_{600}$ was 4 ± 0.5 (yielding a total of >16,000 $OD_{600}$ per replicate).

Vacuole immunoisolation was performed following the MemPrep method (Reinhard et al., 2023, 2024). Briefly, cells were mechanically fragmented with zirconia glass beads using a bead beater, following a differential centrifugation procedure at 3,234 g, 12,000 g, and 100,000 g to remove cell debris and enriching the microsomal fraction, including the vacuole membranes, which were labeled with a bait tag (Myc-3C-3xFLAG) attached to the Mam3 protein. Then, controlled pulses of sonication separated clumps of vesicles and produced smaller microsomes for immunoisolation. Anti-FLAG–coated magnetic beads bind to Mam3 in vacuole membranes, but not to other membranes. For washing, the magnetic beads were immobilized by a magnet, the buffer with all unbound material was removed, and fresh, urea-containing buffer was added. The beads were serially washed and agitated in the absence of a magnetic field to ensure proper mixing and removal of unbound membrane vesicles. The vacuole membranes were eluted using affinity-purified HRV-3C protease, resulting in purified vacuole membranes for lipidomics. The buffer was exchanged, and the sample was concentrated by high-speed centrifugation (200,000 × g). The resuspended pellet was transferred to fresh microcentrifuge tubes, snap-frozen in liquid nitrogen, and stored at −80°C until lipid extraction and mass spectrometry analysis.

## Lipid extraction for mass spectrometry lipidomics

Mass spectrometry–based lipid analysis was performed by Lipotype GmbH as described previously (Ejsing et al., 2009; Klose et al., 2012). Lipids were extracted using a two-step chloroform/methanol procedure (Ejsing et al., 2009). The samples were spiked with an internal lipid standard mixture containing CDP-DAG 17:0/18:1, cardiolipin 14:0/14:0/14:0/14:0 (CL), ceramide 18:1;2/17:0 (Cer), diacylglycerol 17:0/17:0 (DAG), lysophosphatidate 17:0 (LPA), lysophosphatidylcholine 12:0 (LPC), lysophosphatidylethanolamine 17:1 (LPE), lysophosphatidylglycerol 17:1 (LPG), lysophosphatidylinositol 17:1 (LPI), lysophosphatidylserine 17:1 (LPS), phosphatidate 17:0/14:1 (PA), phosphatidylcholine 17:0/14:1 (PC), phosphatidylethanolamine 17:0/14:1 (PE), phosphatidylglycerol 17:0/14:1 (PG), phosphatidylinositol 17:0/14:1 (PI), phosphatidylserine 17:0/14:1 (PS), ergosterol ester 13:0 (EE), triacylglycerol 17:0/17:0/17:0 (TAG), stigmastatrienol, inositolphosphorylceramide 44:0;2 (IPC), mannosylinositolphosphorylceramide 44:0;2 (MIPC), and mannosyl-di-(inositolphosphoryl)ceramide 44:0;2 (M(IP)2C). After extraction, the organic phase was transferred to an infusion plate and dried in a speed vacuum concentrator. The first-step dry extract was resuspended in 7.5 mM ammonium acetate in chloroform/methanol/propanol (1:2:4; V:V:V) and the second-step dry extract in a 33% ethanol solution of methylamine in chloroform/methanol (0.003:5:1; V:V:V). All liquid handling steps were performed using the Hamilton Robotics STARlet robotic platform with the Anti-Droplet Control feature for organic solvent pipetting.

## Mass spectrometry

Samples were analyzed by direct infusion on a Q Exactive mass spectrometer (Thermo Fisher Scientific) equipped with a TriVersa NanoMate ion source (Advion Biosciences). Samples were analyzed in both positive and negative ion modes with a resolution of Rm/z = 200 = 280,000 for MS and Rm/z = 200 = 17,500 for MS/MS experiments, in a single acquisition. MS/MS was triggered by an inclusion list encompassing the corresponding MS mass ranges scanned in 1 Da increments (Surma et al., 2015). MS and MS/MS data were combined to monitor EE, DAG, and TAG ions as ammonium adducts; PC as an acetate adduct; and CL, PA, PE, PG, PI, and PS as deprotonated anions. MS was used only to monitor CDP-DAG, LPA, LPE, LPG, LPI, LPS, IPC, MIPC, and M(IP)2C as deprotonated anions; Cer and LPC as acetate adducts; and ergosterol as a protonated ion of an acetylated derivative (Liebisch et al., 2006).

## Data analysis and postprocessing

Data were analyzed with in-house–developed lipid identification software based on LipidXplorer (Herzog et al., 2012). Data

postprocessing and normalization were performed using an in-house–developed data management system. Only lipid identifications with a signal-to-noise ratio >5, and a signal intensity fivefold higher than in the corresponding blank samples were considered for further data analysis. These data were used to create final graphs and statistical analyses using GraphPad.

## Molecular dynamics simulations

### System setup

The structure of the DysF domain (residues 284–408) of Pex30 (UniProt accession number: Q06169) was predicted using ColabFold (Mirdita et al., 2022). The prediction was highly accurate with a predicted local distance difference test score of 95.5 (Jumper et al., 2021). The atomistic DysF structure was converted to CG using the martinize script (Kroon et al., 2024). An elastic network with a force constant of 500 kJ mol$^{-1}$ nm$^{-2}$ was applied to retain the secondary structure of the domain, with upper and lower elastic bond cutoffs of 0.9 and 0.5 nm, respectively. The insane Python script was used to generate the CG membrane bilayer and to place the CG DysF domain at least 2.5 nm away from the membrane in the z direction (Wassenaar et al., 2015). The final box size was of 17 × 17 × 25 nm in x, y and z directions, respectively. The box was then solvated with water and a NaCl concentration of 0.12 M. A total of seven membrane systems were simulated with the DysF domain monomer, with the following lipid compositions: 100% dioleoyl-phosphatidylcholine (DOPC) system; four systems containing DOPC and dioleoyl-phosphatidic-acid (DOPA) in increments of 10% DOPA from 10 to 40%; two systems where the 40% DOPA had been substituted with 40% dioleoyl-phosphatidylserine (DOPS) or 40% dioleoyl-phosphatidylethanolamine (DOPE).

### Molecular dynamics simulations

All CG-MD simulations were performed with the GROMACS package (v 2023.3) (Van Der Spoel et al., 2005) using the Martini 3 force field (Souza et al., 2021). All systems were initially minimized using a steepest descent algorithm. NPT equilibration was run for 250 ps with restraints on the protein backbone before production runs. For all systems, simulations were kept at 310K using a velocity-rescale thermostat (Bussi et al., 2007) that separately coupled the protein, membrane, and solvent. The Parrinello–Rahman barostat (Parrinello and Rahman, 1981) was used to maintain the pressure constant at 1 bar using a semi-isotropic pressure coupling scheme. To calculate the nonbonded interactions, a Verlet scheme with a buffer tolerance of 0.005 was used. Reaction-field electrostatics was used to compute the coulombic interactions, while the cutoff method was used for the van der Waals terms. Both former terms have a cutoff distance of 1.1 nm and follow the Verlet cutoff scheme for the potential shift (de Jong et al., 2016). A time step of 20 fs was used with the md integrator. Eight independent replicas of 4 μs each were simulated for each system.

For AA simulations, Mstool (Kim, 2023) was first used to backmap a randomly selected frame with the protein bound to the respective membranes from CG into AA. The CHARMM36 force field (Lee et al., 2016) was used in combination with the GROMACS (v 2023.3) package (Van Der Spoel et al., 2005). The

AA systems were initially minimized to 5,000 steps. Next, two equilibrations in the NVT ensemble were run for 125 ps, followed by four equilibrations in the NPT ensemble. With each iteration of the NPT equilibrations, constraints on the DysF backbone and lipid bilayer were gradually removed until the entire system was allowed to move freely (Jo et al., 2008). For the production runs, a time step of 2 fs was used with the md integrator. Three independent replicas were simulated for 200 each. Temperature was kept at 310K using a Nosé–Hoover thermostat (Nosé, 1984), while pressure was maintained at 1 bar using a semi-isotropic Parrinello–Rahman barostat (Parrinello and Rahman, 1981). A Verlet cutoff scheme with a cutoff value of 1.2 nm was used to calculate van der Waals and coulombic interactions. Beyond 1.2 ns, particle mesh Ewald was used to compute long-range interactions. Hydrogen bonds were constrained using the LINCS algorithm (Hess et al., 1997).

### Simulation analyses

The minimum distance between the protein and the membrane throughout the trajectory was computed using GROMACS's *gmx mindist*. Kernel density estimation (kde) was used to assess the membrane-binding events. Acquiring the derivative of the kde defined the bound states of DysF as the instances with minimum distance below or equal to 0.7 nm. The binding percentage was calculated as the area below the curve. Error bars report the standard error as calculated with respect to each individual replica.

Time traces of the minimum distance for each individual residue were used to compute the residue-bound fraction. A residue was considered bound if the minimum distance between any bead of the residue and any bead of the membrane was <0.5 nm. Bound instances for each residue across all replicas were then counted, summed, and normalized over the frames for all replicas. The surface heatmap representation of the binding residues of the DysF domain displays the computed values for the normalized binding frequency of each residue.

The insertion depth analysis for AA-MD simulations was obtained using an adapted mdAnalysis script (Rogers and Geissler, 2023; Gowers et al., 2016; Michaud-Agrawal et al., 2011). All graphical representations were plotted using Matplotlib (Hunter, 2007). All visual representations were created using VMD (Humphrey et al., 1996).

### Statistical analysis

Data for all experiments were generated from at least three independent experiments, and no samples were excluded. For microscopy analysis, cells were randomly selected for analysis and a representative image is shown. For the quantification of the microscopy data, >150 cells per condition and/or genotype were scored from multiple microscopy fields, except when indicated. Distributions are presented as the mean ± SD. Data were tested for normal distribution using D'Agostino & Pearson and Shapiro–Wilk tests and passed at least one of the tests. If the number of points was too small to test normal distribution, data distribution was assumed to be normal. Statistical comparisons were made using the appropriate test for the type of data and are mentioned in the figure legends (GraphPad Prism 10.0.2; ****P < 0.0001; ***P < 0.001; **P < 0.01; *P < 0.05; ns P > 0.5).

## Online supplemental material

Fig. S1 shows further characterization of ER MCS in WT, *pex29Δ*, *pex30Δ*, and *nvj1Δ* cells. Fig. S2 shows the characterization of DysF domains of Pex30 and that its adaptor proteins are not required for complex assembly but affect their localization. Fig. S3 shows that DysF domains from various proteins can bind PA in vitro and the effect of *pex30Δ* mutation in intracellular distribution of various lipid biosensors. Fig. S4 shows the impact of various mutations on the whole-cell and vacuole lipidomes and on the distribution of a PA biosensor localized to the ER. Fig. S5 shows the analysis of Pex30 phosphorylation during exponential and stationary phases and the impact of Pex30 phospho-mutants in Nvj1 localization. Table S1 lists the yeast strains used in this study. Table S2 lists the plasmids used in this study. Table S3 summarizes previously reported Pex30 posttranslational modifications. Videos 1, 3, 5, and 7 correspond to a Z-plan stack of electron microscopy images revealing the morphology of the contact site between the nucleus, vacuole, and LDs in different genetic backgrounds, while Videos 2, 4, 6, and 8 correspond to the 3D reconstruction of the same MCS, respectively.

## Data availability

All data used in this study are available upon request. The raw proteomics dataset generated during this study is available at PRIDE as PXD056055.

## Acknowledgments

We thank S. Martin (University of Geneva, Geneva, Switzerland) for the mCherry-D4H sensor, W. Prinz (UT Southwestern Medical Center, Dallas, TX, USA) for the human PKD(136–343)-GFP-Ubc6TM sensor, and K. Toropova (University of Oxford, Oxford, UK) and C. Melia (University of Oxford) for discussions about EM data analysis.

J.V. Ferreira was supported by an Oxford-EP Abraham Research Fund Graduate Scholarship. P. Carvalho was supported by a Biotechnology and Biological Sciences Research Council grant (BB/W015722/1) and an investigator award from The Wellcome Trust (223153/Z/21/Z). S. Vanni was supported by funding from the European Research Council under the European Union's Horizon 2020 research and innovation program (grant agreement no. 803952) and by grants from the Swiss National Supercomputing Centre under project IDs s1176 and s1269. Open Access funding provided by University of Oxford.

Author contributions: J.V. Ferreira: conceptualization, data curation, formal analysis, investigation, methodology, project administration, resources, validation, visualization, and writing—original draft, review, and editing. Y. Ahmed: data curation, formal analysis, investigation, methodology, validation, visualization, and writing—original draft, review, and editing. T. Heunis: formal analysis, investigation, visualization, and writing—review and editing. A. Jain: investigation. E. Johnson: investigation, methodology, and writing—review and editing. M. Räschle: data curation, formal analysis, investigation, and writing—review and editing. R. Ernst: methodology, resources, supervision, validation, and writing—review and editing. S. Vanni: conceptualization, funding acquisition, methodology, project administration, resources, supervision, validation, and writing—original draft, review, and editing. P. Carvalho: conceptualization, funding acquisition, investigation, project administration, resources, supervision, and writing—original draft, review, and editing.

Disclosures: The authors declare no competing interests exist.

Submitted: 6 September 2024

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

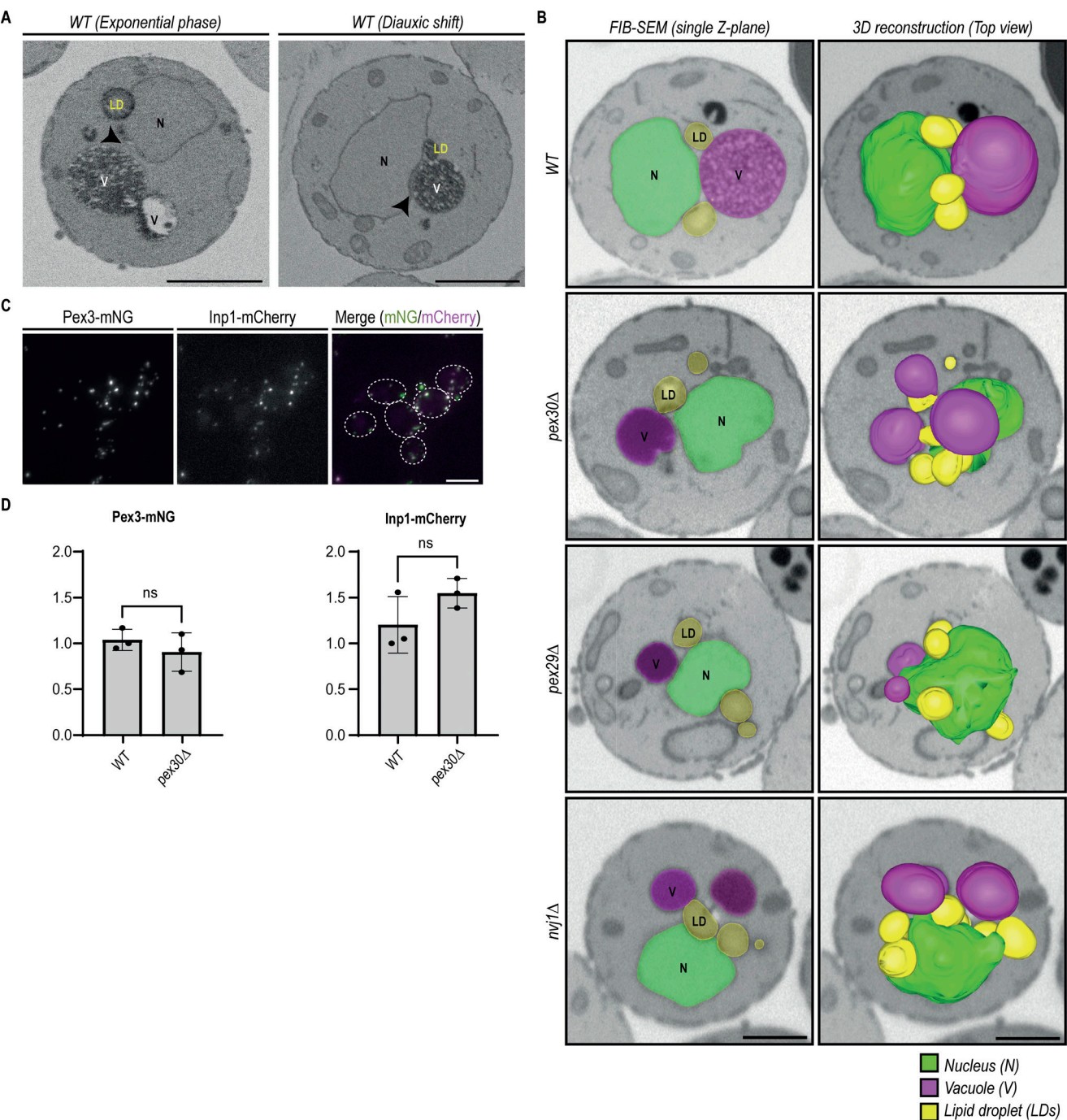

Figure S1.   **Pex30 complexes contribute to MCS formation. (A)** Single Z-slices of *WT* spheroplasts during the exponential phase and diauxic shift from volumes acquired using SBF-SEM 3View. The site of the minimal distance between the nucleus and the vacuole is indicated by an arrowhead. N, nucleus; V, vacuole; LD, lipid droplet; SBF-SEM, serial block-face scanning electron microscopy. Bars, 1 µm. **(B)** Single Z-slices (left) and volume reconstruction (right) of *WT*, *pex30Δ*, *pex29Δ*, and *nvj1Δ* spheroplasts during the diauxic shift acquired using FIB-SEM. The ER is reconstructed in green, the vacuole in magenta, and LDs in yellow. Bar, 1 µm. **(C)** Localization of endogenous Pex3-mNG and Inp1-mCherry in exponential growing *WT* cells. Bar, 5 µm. **(D)** Quantification of Pex3 and Inp1 protein levels in cells in the indicated genotype. Three independent experiments were conducted, and Student's *t* test (two-tailed) was performed to compare the normalized intensity between conditions (ns, P > 0.5). Bars represent the SD. FIB-SEM, focused ion beam scanning electron microscopy.

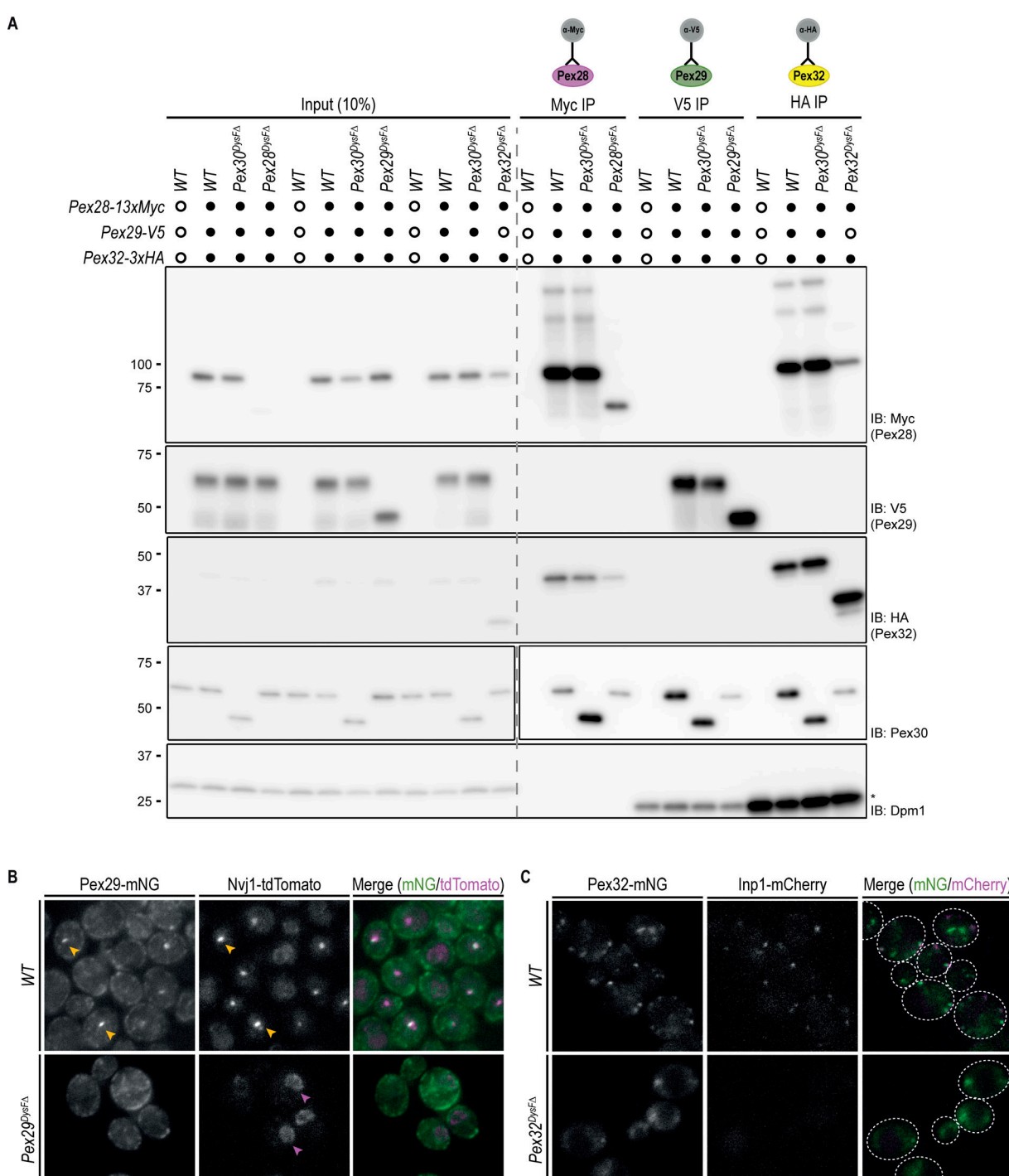

**Figure S2.** **DysF domains of Pex30 adaptors are also required for complex localization. (A)** Crude membrane fractions of cells expressing endogenously tagged variants of Pex28-13xMyc, Pex29-V5, and Pex32-3xHA, or untagged proteins as control, were solubilized with detergent, and the extracts were subjected to immunoprecipitation (IP) with anti-Myc, V5, or HA antibodies. Eluted proteins were separated by SDS-PAGE and analyzed by western blotting. Pex28-Myc, Pex29-V5, Pex30, and Pex32-HA were detected with anti-Myc, anti-V5, anti-Pex30, and anti-HA, respectively. Dpm1, used as a loading control, was detected with anti-Dpm1 antibody. *, IgG light chain. IB, immunoblot. The position of molecular weight markers (in kDa) is indicated. **(B)** Localization of endogenous Pex29-mNG variants and Nvj1-tdTomato in cells during the diauxic shift. Yellow arrowheads highlight colocalization between Pex29 and Nvj1, and magenta arrowheads highlight the non-concentrated Nvj1. Bar, 5 μm. **(C)** Localization of endogenous Pex32-mNG variants and Inp1-mCherry in exponentially growing cells. Bar, 5 μm. Source data are available for this figure: SourceData FS2.

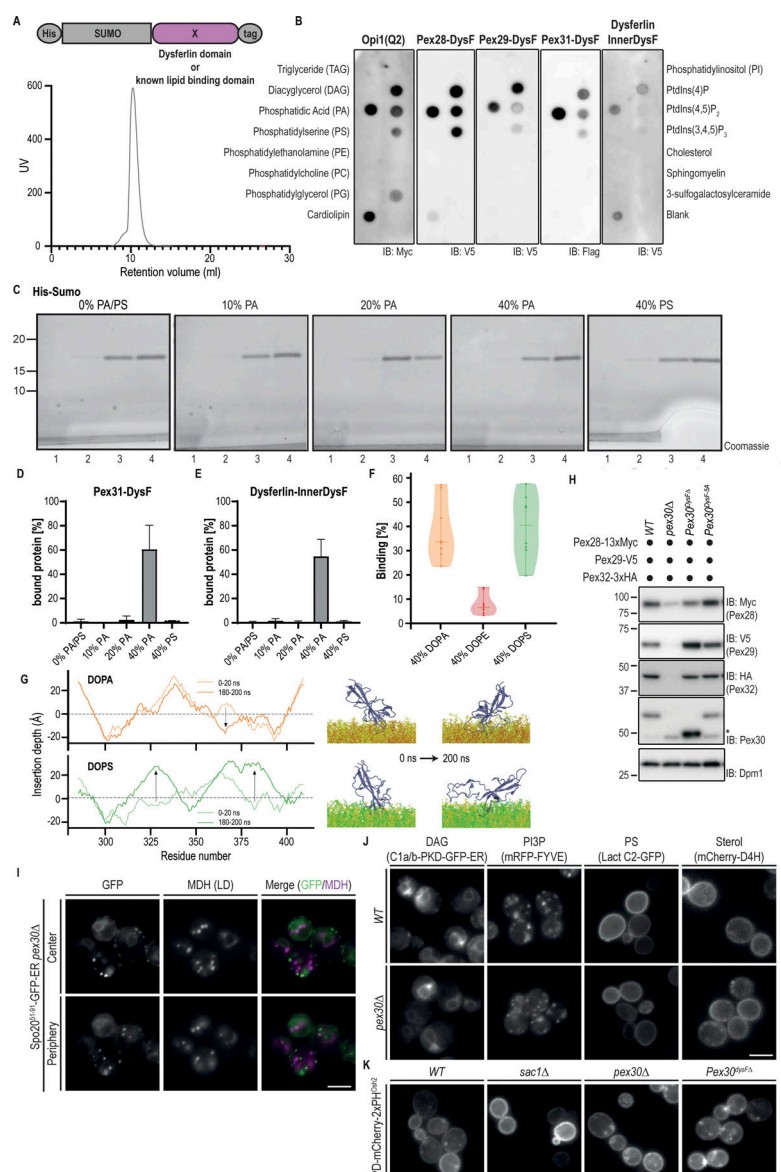

Figure S3.   **DysF domains bind PA in vitro. (A)** DysF and Opi1-PA-binding domains were expressed as fusion proteins to the epitope tags indicated and were detected with antibodies against the C-terminal tag. Bottom: size-exclusion chromatogram of the His-SUMO-Pex30-DysF-HA purification run. UV in arbitrary units represents the amount of protein based on the absorbance at 280 nm. **(B)** Both Opi1 and DysF domains bind to PA and more weakly to phosphoinositides. Purified proteins were incubated with the indicated lipids immobilized in a nitrocellulose membrane. Proteins were expressed as a fusion protein to the epitope tags indicated and were detected with antibodies against their C-terminal tag. IB, immunoblot. **(C)** Liposome flotation assay of His-Sumo as described in Fig. 3 B using liposomes containing different PA or PS concentrations. On the bottom of the top fraction (1), the lipids from the liposomes can be observed. The position of molecular weight markers (in kDa) is indicated. **(D)** Quantification of the percentage of DysF from Pex31 cofractionating with liposomes to the top fraction of experiments described as in Fig. 3 B. The bars represent the SD. **(E)** Quantification of the percentage of InnerDysF from human DysF cofractionating with liposomes to the top fraction of experiments described as in Fig. 3 B. The bars represent the SD. **(F)** Binding of the DysF domain of Pex30 to membrane systems with 40% of DOPA, DOPE, or DOPS in CG-MD simulations. **(G)** Insertion depth (Å) per residue of Pex30-DysF in PA-rich (top, orange) and PS-rich (green) membranes during the first and last 20 ns of AA-MD simulations. The dotted line denotes the surface of the membrane, while arrows indicate main variations in insertion depth over time. Blue, DysF; yellow, DOPC; orange, DOPA, and green, DOPS. **(H)** Steady-state levels of endogenously tagged Pex28-13xMyc, Pex29-V5, and Pex32-3xHA in cells with the indicated genotype. Whole-cell extracts were prepared from exponentially growing cells, separated by SDS-PAGE, and analyzed by western blotting. Pex28-13xMyc, Pex29-V5, Pex30, and Pex32-3xHA were detected with anti-Myc, anti-V5, anti-Pex30, and anti-HA, respectively. Dpm1, used as a loading control, was detected with anti-Dpm1 antibody. Pex30[DysF-5A] corresponds to a variant of Pex30 with alanine mutations on W298, I301, K304, F392, and Y395. *, not specific band. IB, immunoblot. The position of molecular weight markers (in kDa) is indicated. **(I)** Localization of Spo20[51–91]-GFP-ER in pex30Δ cells. Cells were analyzed in the diauxic shift after overnight growth in SC medium, and LDs were stained with the neutral lipid dye MDH. Individual Z-planes corresponding to the center and the periphery of the cell are shown. Bar, 5 μm. **(J)** Distribution of lipid sensors for DAG (Ca1/b-PKD-GFP-ER), PI3P (mRFP-FYVE), PS (Lact-C2-GFP), and sterol (mCherry-D4H) in WT and pex30Δ cells during the exponential phase after overnight growth in SC medium. Bar, 5 μm. **(K)** Distribution of the PI(4)P lipid sensor GPD-mCherry-2xPH[Osh2] in cells of the indicated genotype during the exponential phase after overnight growth in SC medium. As a control, sac1Δ cells, lacking the phosphoinositide phosphatase Sac1, were analyzed. Bar, 5 μm. Source data are available for this figure: SourceData FS3.

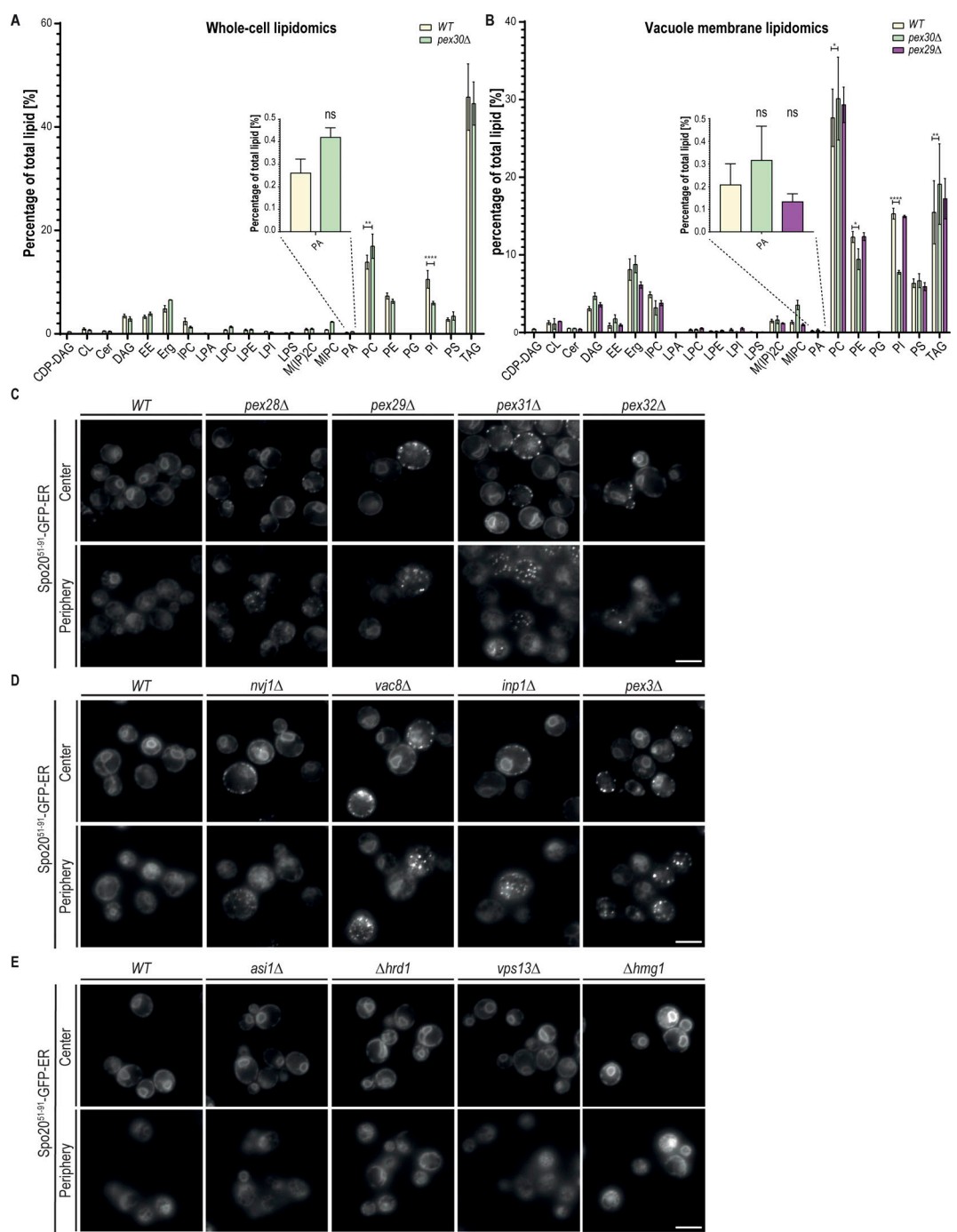

Figure S4. **MCS regulated by Pex30 complexes contribute to normal PA distribution. (A)** Percentage of the membrane lipid type in whole-cell lysates from cells in the stationary growth phase with the indicated genotype. The bars represent the mean and SD. One-way ANOVA and Dunnett's multiple comparisons were performed to compare the percentage of each lipid type with the WT condition (****P < 0.0001; **P < 0.01; ns, not significant). (CDP-DAG: Cytidine Diphosphate Diacylglycerol; CL: Cardiolipin; Cer: Ceramide; DAG: Diacylglycerol; EE: Sterol Ester; Erg: Ergosterol; IPC: Inositol Phosphorylceramide; LPA: Lyso-Phosphatidic Acid; LPC: Lysophosphatidylcholine; LPE: Lysophosphatidylethanolamine; LPI: Lysophosphatidylinositol; LPS: Lysophosphatidylserine; M(IP)2C: Mannosyl di-(inositolphosphoryl)-ceramide; MIPC: Mannosyl inositolphosphoryl-ceramide; PA: Phosphatidic Acid; PC: Phosphatidylcholine; PE: Phosphatidylethanolamine; PG: Phosphatidylglycerol; PI: Phosphatidylinositol; PS: Phosphatidylserine; TAG: Triacylglycerol). **(B)** Same as A but in membranes from purified vacuoles from cells with the indicated genotype. The bars represent the mean and SD. One-way ANOVA and Dunnett's multiple comparisons were performed to compare the percentage of each lipid type with the WT condition (****P < 0.0001; **P < 0.01; *P < 0.05; ns, not significant). **(C)** Localization of Spo20[51–91]-GFP-ER in cells with mutations in Pex30 family members. Cells were analyzed during the diauxic shift after overnight growth in SC medium. Individual Z-planes corresponding to the center and the periphery of the cell are shown. Bar, 5 μm. **(D)** Localization of Spo20[51–91]-GFP-ER in cells with mutations on the tether proteins of ER–peroxisome MCS and NVJ. Cells were analyzed during the diauxic shift after overnight growth in SC medium. Individual Z-planes corresponding to the center and the periphery of the cell are shown. Bar, 5 μm. **(E)** Localization of Spo20[51–91]-GFP-ER in cells with mutations not related to Pex30-related MCS. Cells were analyzed during the diauxic shift after overnight growth in SC medium. Individual Z-planes corresponding to the center and the periphery of the cell are shown. Bar, 5 μm.

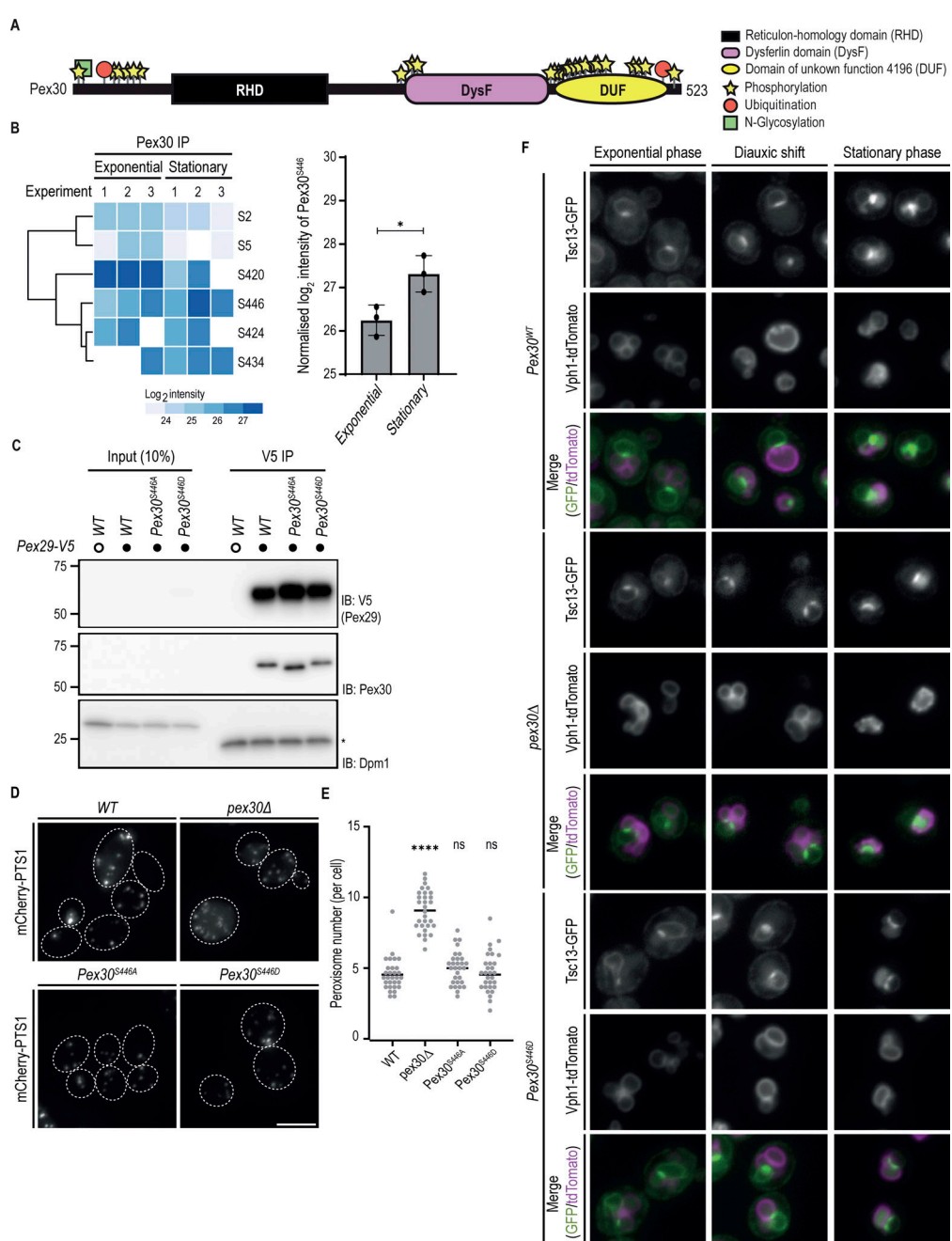

Figure S5. **Pex30-S446 is highly phosphorylated upon the diauxic shift. (A)** Distribution of the posttranslational modifications previously reported for Pex30. Check supplementary data Table S3 for details on the residues and respective studies. **(B)** Pex30-S446 is more phosphorylated during the stationary phase. Log$_2$ intensity of peptides phosphorylated in distinct Pex30 residues during exponential and stationary growth phases. White represents no detection of the peptide in the sample. Right: the intensity of peptides that contained phosphorylated S446 was normalized to the amount of total Pex30 protein in the corresponding sample. The bars represent the mean and SD. Three independent experiments were conducted, and Student's $t$ test (two-tailed) was performed to compare the normalized intensity between conditions (*P < 0.05). **(C)** Pex29 interacts with Pex30 independently of its phosphorylation status. Crude membrane fractions of cells with the indicated genotypes and expressing endogenous Pex29-V5, or untagged proteins as control, were solubilized with detergent, and extracts were subjected to IP with anti-V5 antibody. Eluted proteins were separated by SDS-PAGE and analyzed by western blotting. Pex29-V5 and Pex30 were detected with anti-V5 and anti-Pex30 antibody. Dpm1, used as a loading control, was detected with anti-Dpm1 antibody. *, IgG light chain. IB, immunoblot; IP, immunoprecipitation. The position of molecular weight markers (in kDa) is indicated. **(D)** Distribution of peroxisomes in cells with the indicated genotype during exponential growth. Peroxisomes were labeled by the mCherry-PTS1 marker. Please note the increase of cytosolic fluorescence in the mutant cells, corresponding to non-imported mCherry-PTS1. Images correspond to maximum intensity Z-projections. Bar, 5 μm. **(E)** Quantification of the number of peroxisomes per cell, in cells grown as in D. Three independent experiments were analyzed (>30 cells/genotype/experiment were counted). Each dot corresponds to a cell, and the bars represent the mean and SD. Ordinary one-way ANOVA and Dunnett's multiple comparisons were performed to compare the number of peroxisomes between mutants and the WT condition (****, P < 0.0001; ns, not significant). **(F)** Localization of the NVJ component Tsc13 was analyzed in cells with the indicated genotype during the exponential, diauxic shift, and stationary phases. Endogenous Tsc13 was tagged with GFP (Tsc13-GFP), and endogenously expressed Vph1-tdTomato was used as a vacuole marker. Bar, 5 μm. Source data are available for this figure: SourceData FS5.

Video 1.   **Morphology of MCS between the nucleus, vacuole, and LDs in a *WT* cell during the diauxic shift, shown in** Fig. 1 F**.** Pixel resolution: 9 × 9 × 20 nm.

Video 2.   **3D reconstruction of MCS between the nucleus, vacuole, and LDs in a *WT* cell during the diauxic shift, shown in** Video 1**.** The ER is shown in green, the vacuole in purple, and LDs in yellow.

Video 3.   **Morphology of the nucleus, vacuole, and LDs in a *pex30Δ* cell during the diauxic shift, shown in** Fig. 1 F**.** Pixel resolution: 9 × 9 × 20 nm.

Video 4.   **3D reconstruction of the nucleus, vacuole, and LDs in a *pex30Δ* cell during the diauxic shift, shown in** Video 3**.** The ER is shown in green, the vacuole in purple, and LDs in yellow.

Video 5.   **Morphology of the nucleus, vacuole, and LDs in a *pex29Δ* cell during the diauxic shift, shown in** Fig. 1 F**.** Pixel resolution: 9 × 9 × 20 nm.

Video 6.   **3D reconstruction of the nucleus, vacuole, and LDs in a *pex29Δ* cell during the diauxic shift, shown in** Video 5**.** The ER is shown in green, the vacuole in purple, and LDs in yellow.

Video 7.   **Morphology of the nucleus, vacuole, and LDs in an *nvj1Δ* cell during the diauxic shift, shown in** Fig. 1 F**.** Pixel resolution: 9 × 9 × 20 nm.

Video 8.   **3D reconstruction of the nucleus, vacuole, and LDs in an *nvj1Δ* cell during the diauxic shift, shown in** Video 7**.** The ER is shown in green, the vacuole in purple, and LDs in yellow.

**Provided online are Table S1, Table S2, and Table S3. Table S1 shows yeast strains used in this study. Table S2 shows plasmids used in this study. Table S3 shows posttranslational modifications reported for Pex30.**

