## [Peer Review File · The Journal of Cell Biology]

Pex30-dependent membrane contacts sites maintain ER lipid homeostasis

Joana Veríssimo Ferreira, Yara Ahmed, Tiaan Heunis, Aamna Jain, Errin Johnson, Markus Räschle, Robert Ernst, Stefano Vanni, and Pedro Carvalho

Corresponding Author(s): Pedro Carvalho, University of Oxford

Review Timeline:

Submission Date:	2024-09-06
Editorial Decision:	2024-10-17
Revision Received:	2025-01-28
Editorial Decision:	2025-02-24
Revision Received:	2025-03-07

Monitoring Editor: Laura Lackner

Scientific Editor: Andrea Marat

Transaction Report:

DOI: <https://doi.org/10.1083/jcb.202409039>

October 17, 2024

Re: JCB manuscript #202409039

Dr. Pedro Carvalho
University of Oxford
Sir William Dunn School of Pathology
South Parks road
South Parks Road
Oxford, UK OX1 3RE
United Kingdom

Dear Dr. Carvalho,

Thank you for submitting your manuscript entitled "Pex30-dependent membrane contacts sites maintain ER lipid homeostasis". The manuscript was assessed by expert reviewers, whose comments are appended to this letter. We invite you to submit a revision if you can address the reviewers' key concerns, as outlined here.

As you will see in the reviewer comments, two experts in the field agree that this is an important and interesting study that advances our understanding of the molecular determinants that allow Pex30 to regulate multiple membrane contact sites (MCSs). It is pointed out that the work is of high quality. That being said, the reviewers do have a few concerns regarding some of the data and provide clear recommendations to further strengthen these data that should be experimentally addressed in the revision. Specifically, both reviewers raise concerns related to PA and PI4P binding (R1 #1 and 2 and R2 #4) that need to be addressed. In addition, R1's suggestion to further dissect Pex30's role in NVJ size and how this may affect the recruitment of proteins to the MCS (R1 #3) should be addressed. R2 suggests additional controls that should be included in a revision and raises a few other minor points that can likely be addressed with changes to the text. If the question raised by R2 regarding differences in PA species (R2 #6) can be addressed without additional experimentation, those data would be a very interesting addition to the study. However, we would consider the need to perform additional lipidomic analysis beyond the scope of what is required for a revision.

GENERAL GUIDELINES:

Text limits: Character count for an Article is < 40,000, not including spaces. Count includes title page, abstract, introduction, results, discussion, and acknowledgments. Count does not include materials and methods, figure legends, references, tables, or supplemental legends.

Figures: Articles may have up to 10 main text figures. Figures must be prepared according to the policies outlined in our Instructions to Authors, under Data Presentation, <https://jcb.rupress.org/site/misc/ifora.xhtml>. All figures in accepted manuscripts will be screened prior to publication.

Supplemental information: There are strict limits on the allowable amount of supplemental data. Articles may have up to 5 supplemental figures. Up to 10 supplemental videos or flash animations are allowed. A summary of all supplemental material should appear at the end of the Materials and methods section.

Please note that JCB now requires authors to submit Source Data used to generate figures containing gels and Western blots with all revised manuscripts. This Source Data consists of fully uncropped and unprocessed images for each gel/blot displayed in the main and supplemental figures. Since your paper includes cropped gel and/or blot images, please be sure to provide one Source Data file for each figure that contains gels and/or blots along with your revised manuscript files. File names for Source Data figures should be alphanumeric without any spaces or special characters (i.e., SourceDataF#, where F# refers to the associated main figure number or SourceDataFS# for those associated with Supplementary figures). The lanes of the gels/blots should be labeled as they are in the associated figure, the place where cropping was applied should be marked (with a box), and molecular weight/size standards should be labeled wherever possible. Source Data files will be made available to reviewers during evaluation of revised manuscripts and, if your paper is eventually published in JCB, the files will be directly linked to specific figures in the published article.

The typical timeframe for revisions is three to four months. If you anticipate any difficulties in meeting this aforementioned revision time limit, please contact us and we can work with you to find an appropriate time frame for resubmission. Please note that papers are generally considered through only one revision cycle, so any revised manuscript will likely be either accepted or rejected.

Thank you for this interesting contribution to Journal of Cell Biology. You can contact us at the journal office with any questions at cellbio@rockefeller.edu.

Sincerely,

Laura Lackner, PhD
Monitoring Editor

Andrea L. Marat, PhD
Deputy Editor

Journal of Cell Biology

Reviewer #1 (Comments to the Authors (Required)):

This study investigates the function of Pex30 and Pex family proteins at yeast membrane contact sites and lipid metabolism. They show that Pex30-Pex29 complexes influence the NVJ contact site. Using volumetric EM, they show yeast lacking Pex30 or Pex29 show more distance between the nucleus and vacuole surface. ER-peroxisome contact sites are also perturbed in pex30ko yeast. Structure function analysis suggests the DUF domain is dispensable for Pex30's role at ER-peroxisome contacts, whereas the Dysferlin (Dys) domain is required. Pex30 required both domains to localize to the NVJ. In vitro biochemical data suggests the Dys domain is a PA binding domain. In vivo, Pex30 cells display changes in PA spatial distribution and the appearance of ER localized PA foci. The study also identified phospho sites on the Pex30 DUF domain that mediates NVJ localization. Pex30 S446D mutants show increased

This is an interesting and potentially important study that adds valuable new data to a still poorly understood protein family that sits at the interface between lipid metabolism and organelle contacts. This study addresses how Pex30 influences NVJ morphology and its lipid binding properties. Identifying phospho-sites that influence Pex30 NVJ recruitment is also important. Two general concerns are that the PA binding assays rely on non-physiological levels of PA, and that more work could be done to dissect Pex30's role in NVJ morphology as it is distinct from Pex30 recruitment. The PA binding experiments remain the weakest aspect of the work, and additional experiments are needed to better understand physiological circumstances where Pex30 and PA interact. Addressing these issues will add to this interesting study.

General thoughts

1_ The Dysferlin domain is proposed to bind PA in vitro, but the concentration of PA on liposomes (40%) is extremely high. PA is a non-bilayer lipid and 40% concentration in liposomes may fragment or perturb liposome integrity, making the assay a bit perplexing. Furthermore, it is unlikely that PA would reach such a high local concentration in biological membranes. Some additional work suggesting that Dysferlin domain binds PA in more physiological regimes would be necessary.

2_Related to point 1, the molecular simulations support the model where the Dysferlin domain binds PA. PS is simulated and no interaction detected. However, the dot blot assay suggests there is binding to PI4P in vitro. If PI4P is modeled in simulation does the Dysferlin domain bind it?

3_Figure 6: the identification of phospho-sites on Pex30 that influence its targeting to the NVJ are interesting and important. However, the study also suggests that pex30 loss may impact NVJ size which could also impact how proteins are recruited to the MCS. From Figure 6A it appears that yeast expressing Pex30 S446D may have brighter or more elongated Nvj1 NVJs.

Larger NVJs may simply recruit proteins non-specifically since they offer more surface area, so controlling for NVJ size in these experiments is important.

Reviewer #2 (Comments to the Authors (Required)):

The work by Ferreira and colleagues investigates the molecular determinants that allow the ER membrane protein Pex30 to regulate multiple MCS in *S. cerevisiae*. The authors provide evidence supporting a role of the dysferlin (DysF) domain of Pex30 at MCS through binding to PA. In addition, they show that the DUF domain is specifically required at the NVJ and is regulated by phosphorylation.

The work is of high quality and rich in state-of-the-art techniques.

I have few points of concern, mostly related with controls for some experiments.

- 1- Fig 1F shows that Inp1 levels in pex30delta cells are comparable to that of wt. Are the levels of Pex3 (which binds Inp1) in pex30 delta also comparable to wt?
- 2- Fig 2C shows that lack of DysF domain results in increased number of peroxisomes. This phenotype has been previously shown by Deori et al (doi: 10.1007/s12013-022-01122-z) and it should be acknowledged.
- 3- For all lipid overlays in Fig 3 and S3 a His-SUMO-HA control protein should be purified in the same manner and used at the same concentration.
- 4- I am not convinced that the DysF domain does not bind PI4P in addition to PA. Could PI4P be tested in simulations instead of PA? BTW, 40% PA is quite high.
- 5- What is the rationale for trapping the Spo20-PA probe in the ER? Why not using a soluble Spo20 or Opi1-based probes? An ER-DAG sensor (doi: 10.1083/jcb.201910177) should be used as control for experiments in Fig5 and S4.
- 6- Did the authors look at levels of different PA species in the lipidomics analysis? For those lipids showing significant changes in pex30delta cells are there any specific species (acyl tails) common to all?
- 7- Minor typo, Echolen should be Echelon in M&M section.

Response to the reviewers

Reviewer 1:

1 – “The Dysferlin domain is proposed to bind PA *in vitro*, but the concentration of PA on liposomes (40%) is extremely high. PA is a non-bilayer lipid and 40% concentration in liposomes may fragment or perturb liposome integrity, making the assay a bit perplexing. Furthermore, it is unlikely that PA would reach such a high local concentration in biological membranes. Some additional work suggesting that Dysferlin domain binds PA in more physiological regimes would be necessary.”

We agree with the reviewer that the concentration of PA used in the liposomes (40%) is high and unlikely to reflect typical levels in biological membranes. However, while liposome flotation is the gold standard assay for assessing lipid binding, it has limitations and does not fully recapitulate the *in vivo* situation.

One important difference is that in the *in vitro* assay, the DysF domain is a soluble protein and the binding to liposomes occurs as a three-dimensional reaction. This contrasts with the *in vivo* scenario, where DysF domains are part of integral membrane proteins, positioned adjacent to the membrane and restricted to two-dimensional diffusion. Therefore, even assuming that DysF has relatively low affinity for PA, binding is likely to occur at much lower PA concentrations *in vivo*.

Another aspect to take into consideration is that Pex30 family members function as oligomers, as shown in our previous paper. The presence of multiple DysF domains within a Pex30 complex likely increases the probability of PA binding. We have now changed the text and included these points in the discussion.

Despite the limitations, the *in vitro* and *in silico* experiments were important to reveal the binding specificity of the DysF to PA. In relation to the PA concentration used in the liposomes, it is common to use PA concentration within this range (40%). Previous studies characterizing the PA-binding activity of proteins such as Opi1 (Loewen et al, 2004; Hofbauer et al, 2018) and Spo20 (Nakanishi et al, 2004) used PA concentrations within a similar range or even higher (up to 60%). At 40% PA we did not notice any changes in the flotation efficiency. One would expect that liposome fragmentation would result in less buoyancy.

In summary, we believe that the *in vitro* and *in silico* experiments provide strong evidence for a specific binding of DysF domain to PA. Moreover, these data agree and further support our *in vivo* results implicating the DysF in PA homeostasis.

2 – “Related to point 1, the molecular simulations support the model where the Dysferlin domain binds PA. PS is simulated and no interaction detected. However, the dot blot assay suggests there is binding to PI4P *in vitro*. If PI4P is modelled in simulation does the Dysferlin domain bind it?”

As suggested by the reviewer, we analyzed further the potential binding of the DysF domain to PI4P. Using the liposome floatation assay, we were unable to detect binding of DysF to PI4P-containing liposomes (5%PI4P) (Figure 3C). Coarse grain MD simulation showed some binding of DysF domain to PI4P containing-membranes, but this interaction was weaker in comparison to PA-containing membranes (see Figure R1).

Figure R1. Comparison of the Pex30 Dysferlin domain binding to PA-enriched (40% PA) vs PI4P-enriched (5% PI4P) lipid bilayers.

To assess whether the loss of Pex30 affects PI4P distribution *in vivo*, we utilized a well-established PI4P biosensor. The localization of the PI4P biosensor was indistinguishable between WT and *pex30Δ* cells. In contrast, the distribution of the probe was defective in *sac1Δ* mutants, known to have aberrant PI4P levels (Figure S3K).

Together, these data indicate that the binding of DysF to PI4P in the lipid strips is likely to be an artifact.

3 - “Figure 6: the identification of phospho-sites on Pex30 that influence its targeting to the NVJ are interesting and important. However, the study also suggests that *pex30* loss may impact NVJ size which could also impact how proteins are recruited to the MCS. From Figure 6A it appears that yeast expressing Pex30 S446D may have brighter or more elongated Nvj1 NVJs. Larger NVJs may simply recruit proteins non-specifically since they offer more surface area, so controlling for NVJ size in these experiments is important.”

We agree with the reviewer’s suggestion. To further characterize the NVJ in *pex30Δ* and *pex30^{S446D}* mutants we analyzed the localization of the NVJ component Tsc13. In these Pex30 mutants, endogenous Tsc13 tagged with GFP localized to the NVJ in a manner similar to WT cells. Our analysis did not reveal any noticeable differences in the fluorescent intensity or morphology of Tsc13-GFP at the NVJ. These results suggest that Pex30 mutations specifically affect Nvj1, consistent with our previous findings (Ferreira and Carvalho JCB 2021).

Reviewer 2:

1 - “Fig 1F shows that Inp1 levels in *pex30Δ* cells are comparable to that of wt. Are the levels of Pex3 (which binds Inp1) in *pex30 Δ* also comparable to wt?”

We thank the reviewer for the suggestion. We have now compared the levels of Pex3 between WT and *pex30Δ* cells and did not observe any significant differences. These data are now presented in Figure 1F and quantified in Figure S1D, together with the levels of Inp1.

2 - “Fig 2C shows that lack of DysF domain results in increased number of peroxisomes. This phenotype has been previously shown by Deori et al (doi: 10.1007/s12013-022-01122-z) and it should be acknowledged.”

This has been fixed.

3 - “For all lipid overlays in Fig 3 and S3 a His-SUMO-HA control protein should be purified in the same manner and used at the same concentration.”

All recombinant proteins were expressed, purified and used at the same concentration in the various assays. Both the text and the figures (Figure 3 and S3) have been changed to make this clear.

4 - “I am not convinced that the DysF domain does not bind PI4P in addition to PA. Could PI4P be tested in simulations instead of PA? BTW, 40% PA is quite high.”

Please see our responses to Reviewer #1, points 1 and 2.

5 - “What is the rationale for trapping the Spo20-PA probe in the ER? Why not using a soluble Spo20 or Opi1-based probes? An ER-DAG sensor (doi: 10.1083/jcb.201910177) should be used as control for experiments in Fig5 and S4.”

We thank the reviewer for this comment. Our initial experiments were done with the soluble Spo20 PA biosensor. These showed abnormal distribution of the reporter into foci specifically in *pex30Δ* cells. However, this phenotype was present in a smaller fraction of cells (~18%). To determine the localization of these foci relative to the ER, we required the introduction of an ER marker with a second color, which posed a challenge considering the large number of genetic backgrounds analyzed. To address this, we developed the Spo20-ER sensor, which proved not only more practical for these analyses but also more sensitive in detecting the PA phenotype. This increased sensitivity is likely due to the difference between 2D diffusion (in the ER-bound sensor) and 3D diffusion (in the soluble Spo20 probe).

As suggested by the reviewer, we have tested the ER-DAG sensor (C1a/b-PKD-GFP-ER) and we did not observe any difference in its distribution between *WT* and *pex30Δ* cells (Figure S3J).

6 - “Did the authors look at levels of different PA species in the lipidomics analysis? For those lipids showing significant changes in *pex30delta* cells are there any specific species (acyl tails) common to all?”

We thank the reviewer for the suggestion. Our lipidomic analysis comparing *WT*, *pex30Δ* and *pex29Δ* cells showed that the overall PA levels are similar among all three strains. In terms of PA species, we observed that both *pex30Δ* and *pex29Δ* cells displayed PA with higher levels of palmitic acid (PA C16; C16) and lower levels of oleic acid (PA C16; C18). Although potentially interesting, the nature of these differences was not explored in the current manuscript.

7 - “Minor typo, Echolen should be Echelon in M&M section.”

We thank the reviewer for noticing this. The text has been fixed.

February 24, 2025

RE: JCB Manuscript #202409039R

Pedro Carvalho
University of Oxford

Dear Dr. Carvalho:

Thank you for submitting your revised manuscript entitled "Pex30-dependent membrane contacts sites maintain ER lipid homeostasis". We would be happy to publish your paper in JCB pending final revisions necessary to meet our formatting guidelines (see details below). Please let us know how you would like to proceed with regards to the related study.

A. MANUSCRIPT ORGANIZATION AND FORMATTING:

- 1) Text limits: Character count for Articles is < 40,000, not including spaces. Count includes abstract, introduction, results, discussion, and acknowledgments. Count does not include title page, figure legends, materials and methods, references, tables, or supplemental legends.
- 2) Figures limits: Articles may have up to 10 main text figures.
- 3) Figure formatting: Scale bars must be present on all microscopy images, including inset magnifications. Molecular weight or nucleic acid size markers must be included on all gel electrophoresis. Aspect ratios of images may not be altered.
- 4) Statistical analysis: Error bars on graphic representations of numerical data must be clearly described in the figure legend. The number of independent data points (n) represented in a graph must be indicated in the legend. Statistical methods should be explained in full in the materials and methods. For figures presenting pooled data the statistical measure should be defined in the figure legends. Please also be sure to indicate the statistical tests used in each of your experiments (either in the figure legend itself or in a separate methods section) as well as the parameters of the test (for example, if you ran a t-test, please indicate if it was one- or two-sided, etc.). Also, if you used parametric tests, please indicate if the data distribution was tested for normality (and if so, how). If not, you must state something to the effect that "Data distribution was assumed to be normal but this was not formally tested."
- 5) Abstract and title: The abstract should be no longer than 160 words and should communicate the significance of the paper for a general audience. The title should be less than 100 characters including spaces. Make the title concise but accessible to a general readership.
- 6) Materials and methods: Should be comprehensive and not simply reference a previous publication for details on how an experiment was performed. Please provide full descriptions in the text for readers who may not have access to referenced manuscripts.
- 7) All antibodies, cell lines, animals, and tools used in the manuscript should be described in full, including accession numbers for materials available in a public repository such as the Resource Identification Portal. Please be sure to provide the sequences for all of your primers/oligos and RNAi constructs in the materials and methods. You must also indicate in the methods the source, species, and catalog numbers (where appropriate) for all of your antibodies. Please also indicate the acquisition and quantification methods for immunoblotting/western blots.
- 8) Microscope image acquisition: The following information must be provided about the acquisition and processing of images:
 - a. Make and model of microscope
 - b. Type, magnification, and numerical aperture of the objective lenses
 - c. Temperature
 - d. Imaging medium
 - e. Fluorochromes
 - f. Camera make and model
 - g. Acquisition software
 - h. Any software used for image processing subsequent to data acquisition. Please include details and types of operations involved (e.g., type of deconvolution, 3D reconstitutions, surface or volume rendering, gamma adjustments, etc.).

10) Supplemental materials: There are strict limits on the allowable amount of supplemental data. Articles may have up to 5 supplemental figures. Please also note that tables, like figures, should be provided as individual, editable files. A summary of all supplemental material should appear at the end of the Materials and methods section.

13) ORCID IDs: ORCID IDs are unique identifiers allowing researchers to create a record of their various scholarly contributions in a single place. Please note that ORCID IDs are now *required* for all authors. At resubmission of your final files, please be sure to provide your ORCID ID and those of all co-authors.

Please note that JCB now requires authors to submit Source Data used to generate figures containing gels and Western blots with all revised manuscripts. This Source Data consists of fully uncropped and unprocessed images for each gel/blot displayed in the main and supplemental figures. For assays performed using capillary electrophoresis and/or immunoassay-based detection, authors should instead provide the electropherogram graph(s) for each experiment, plotting fluorescence/chemiluminescence intensity vs. molecular weight/size. Since your paper includes cropped gel and/or blot images, please be sure to provide one Source Data file for each figure gels, blots, and/or capillary electrophoresis assays along with your revised manuscript files. File names for Source Data figures should be alphanumeric without any spaces or special characters (i.e., SourceDataF#, where F# refers to the associated main figure number or SourceDataFS# for those associated with Supplementary figures). For traditional gels and blots, the lanes of the gels/blots should be labeled as they are in the associated figure, the place where cropping was applied should be marked (with a box), and molecular weight/size standards should be labeled wherever possible. For capillary electrophoresis assays, each trace in the graph should be color-coded and labeled to indicate which protein, gene, or sample is being measured (please try to avoid red/green combinations to accommodate our color-blind readers).

Journal of Cell Biology now requires a data availability statement for all research article submissions. These statements will be published in the article directly above the Acknowledgments. The statement should address all data underlying the research presented in the manuscript. Please visit the JCB instructions for authors for guidelines and examples of statements at (<https://rupress.org/jcb/pages/editorial-policies#data-availability-statement>).

B. FINAL FILES:

****It is JCB policy that if requested, original data images must be made available to the editors. Failure to provide original images upon request will result in unavoidable delays in publication. Please ensure that you have access to all original data images prior to final submission.****

****The license to publish form must be signed before your manuscript can be sent to production. A link to the electronic license to publish form will be sent to the corresponding author only. Please take a moment to check your funder requirements before choosing the appropriate license.****

Thank you for your attention to these final processing requirements. Please revise and format the manuscript and upload materials within 7 days. If you need an extension for whatever reason, please let us know and we can work with you to determine a suitable revision period.

Thank you for this interesting contribution, we look forward to publishing your paper in Journal of Cell Biology.

Sincerely,

Laura Lackner, PhD
Monitoring Editor

Andrea L. Marat, PhD
Deputy Editor

Journal of Cell Biology

Reviewer #1 (Comments to the Authors (Required)):

The revised study has addressed the major concerns. Text has been updated to address the caveats of using liposomes with high PA compositions. Further liposome and MD simulations looking at PI4P binding were interrogated. Effects in genetic mutants on NVJ protein localization were also examined more closely.

Reviewer #2 (Comments to the Authors (Required)):

The authors have performed extra experiments and have addressed my concerns satisfactorily. This is a very comprehensive body of work.